# ERFVII action and modulation through oxygen-sensing in *Arabidopsis thaliana*

Agata Zubrycka [1,5], Charlene Dambire[1,5], Laura Dalle Carbonare[1,4,5], Gunjan Sharma [1], Tinne Boeckx[1], Kamal Swarup[1], Craig J. Sturrock[1], Brian S. Atkinson[1], Ranjan Swarup[1], Françoise Corbineau[2], Neil J. Oldham[3] & Michael J. Holdsworth [1]✉

Oxygen is a key signalling component of plant biology, and whilst an oxygen-sensing mechanism was previously described in *Arabidopsis thaliana*, key features of the associated PLANT CYSTEINE OXIDASE (PCO) N-degron pathway and Group VII ETHYLENE RESPONSE FACTOR (ERFVII) transcription factor substrates remain untested or unknown. We demonstrate that *ERFVII*s show non-autonomous activation of root hypoxia tolerance and are essential for root development and survival under oxygen limiting conditions in soil. We determine the combined effects of *ERFVII*s in controlling gene expression and define genetic and environmental components required for proteasome-dependent oxygen-regulated stability of ERFVIIs through the PCO N-degron pathway. Using a plant extract, unexpected amino-terminal cysteine sulphonic acid oxidation level of ERFVIIs was observed, suggesting a requirement for additional enzymatic activity within the pathway. Our results provide a holistic understanding of the properties, functions and readouts of this oxygen-sensing mechanism defined through its role in modulating ERFVII stability.

Oxygen ($O_2$) is a central molecule of eukaryotic metabolism, produced in plants, algae and bacteria through photosynthesis, and required for aerobic respiration and a multitude of biochemical reactions[1,2]. Acute hypoxia (reduced environmental $O_2$ availability) is the central problem of flooding and waterlogging, agriculturally important abiotic stresses that are increasing due to climate change[3]. Hypoxia is also an important component of normal plant development (chronic hypoxia for example, in apical meristems) or induced by pathogens[1,2]. A mechanism for oxygen-sensing in plants was discovered more than 10 years ago[4,5], and is analogous to that of the animal Hypoxia Inducible Factor (HIF) system, including proteasomal destruction of transcription factors following covalent attachment of oxygen via dioxygenase enzymes, though the mechanisms are not related[2]. Work in mammalian systems initially showed that the Arg/N-degron pathway acts as a sensor for both oxygen and nitric oxide (NO)[6,7] Plant oxygen-sensing

was shown to require the $O_2$ (and NO)-sensitive destruction of substrate proteins that initiate with amino-terminal Cysteine (Nt-Cys), including the group VII ETHYLENE RESPONSE FACTOR (ERFVII) transcription factors, VERNALIZATION (VRN)2[8] and LITTLE ZIPPER (ZPR) 2[9], through the PLANT CYSTEINE OXIDASE (PCO) branch of the PROTEOLYSIS (PRT)6 N-degron pathway, hereafter the PCO N-degron pathway (Fig. 1a). Current understanding of the mechanism is that Nt-Cys is revealed in these Met[1]-Cys[2] initiating substrates after cleavage of the methionine by MetAP activity. Following oxidation of Nt-Cys by PCO enzymes, oxidised Cys is arginylated by ARGINYL TRANSFERASEs (ATEs), allowing recognition by the E3 ligase PRT6, ubiquitylation and degradation via the Ubiquitin Proteasome System (UPS). Reduced oxygen availability inhibits PCOs[10] resulting in accumulation of substrates. Methionine excision of substrates has not been shown in plants, though is predicted from the substrate specificity of highly

[1]School of Biosciences, University of Nottingham, LE12 5RD Loughborough, UK. [2]UMR 7622 CNRS-UPMC, Biologie du développement, Institut de Biologie Paris Seine, Sorbonne Université, Paris, France. [3]School of Chemistry, University of Nottingham, University Park, Nottingham NG7 2RD, UK. [4]Present address: Department of Biology, University of Oxford, OX1 3RB Oxford, UK. [5]These author contributed equally: Agata Zubrycka, Charlene Dambire, Laura Dalle Carbonare. ✉e-mail: michael.holdsworth@nottingham.ac.uk

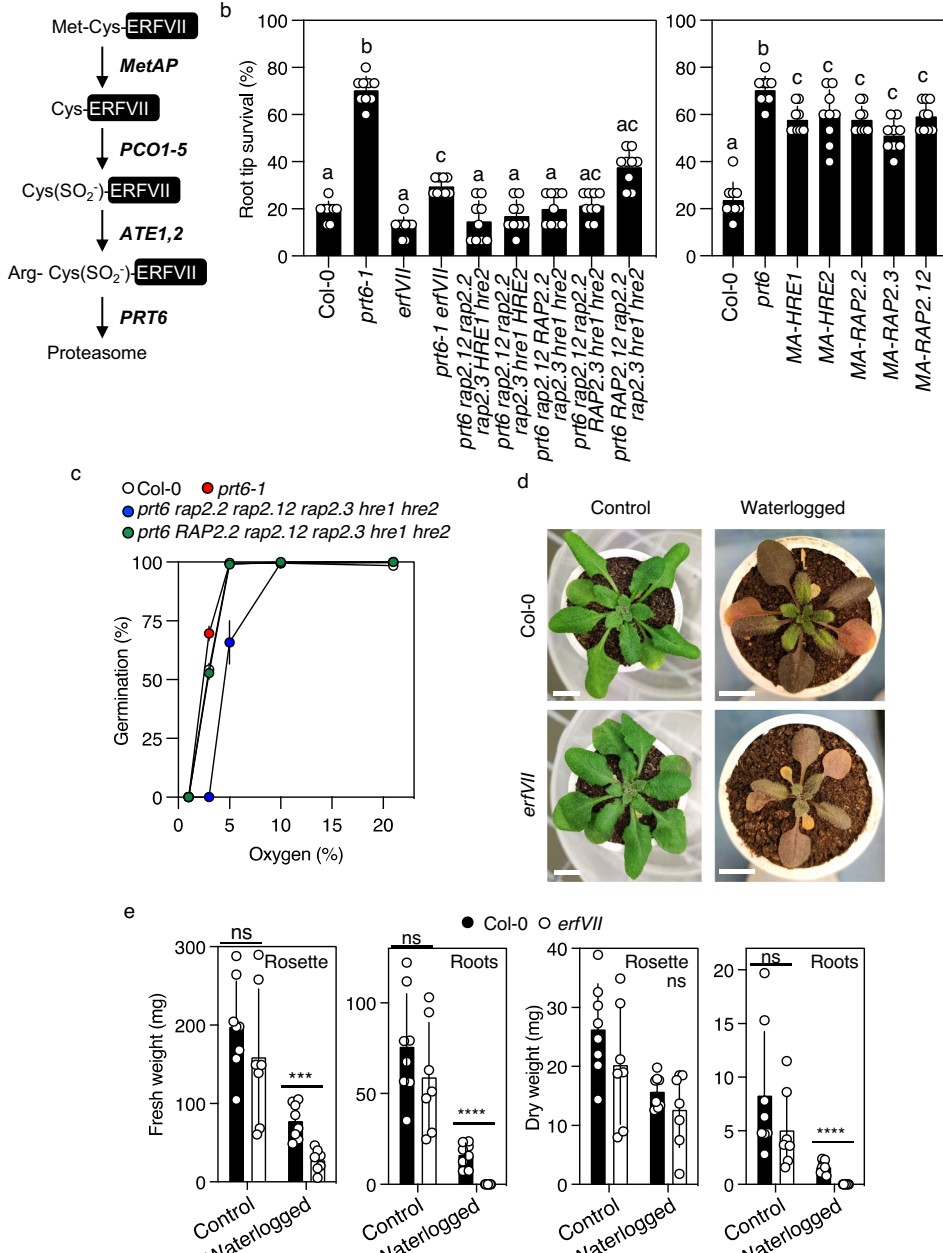

**Fig. 1 | ERFVIIs as determinants of plant hypoxia tolerance. a** Schematic of the currently accepted PCO N-degron pathway. The changing N-terminus of ERFVIIs is shown in relation to enzyme activities. MetAP, METHIONINE AMINOPEPTIDASE; PCO, PLANT CYSTEINE OXIDASE; ATE, ARGINYL TRANSFERASE; PRT6, PROTEO-LYSIS6. The conversion of amino-terminal Cys to Cys-sulfinate ($SO_2^-$) is shown. Residues indicated by three letter code. **b** Root tip survival of wild-type (Col-0) and mutant seedlings subjected to 4 h hypoxia and 3 days recovery. MA-ERFVIIs indicates individual *promERFVII:C²A-ERFVII* transgene in Col-0 background. Data are presented as mean ± SD ($n = 9$) significant differences denoted with letters for one-way ANOVA ($p < 0.05$). **c** Germination under different levels of ambient oxygen for wild-type (Col-0) and mutant seeds. Data are presented as mean ± SD ($n = 3$). **d** Images of four-week-old wild-type (Col-0) or *erfVII* plants grown in sandy-clay loam soil in control conditions or upon waterlogging treatment for 7 days. Scale bar is 1 cm. **e** Weights of rosettes and roots of four-week-old wild-type (Col-0) and *erfVII* mutant with or without waterlogging treatment of the duration of 7 days. Data are presented as mean ± SD ($n = 8$), *t*-test, ***$p$-value < 0.0002 and ****$p$-value < 0.0001, 'ns' not significant.

conserved MetAP activity[11]. Biochemical analysis in vitro using recombinant enzymes purified from *Escherichia coli* showed that PCOs add molecular $O_2$ to Nt-Cys to give Nt-Cys($SO_2^-$) (sulfinate), that can be arginylated by ATEs[12]. PCOs were originally identified in plants[13], and more recently an N-terminal cysteine oxidase equivalent, ADO, was also identified in animals[14]. Although there is an absolute requirement for NO for substrate degradation in vivo through this pathway[6,15], the site of action within the pathway remains unknown[16]. Previously it was shown that the stability of ERFVIIs can be increased genetically by removing components of the PCO N-degron pathway including PCOs[13],

ATE[5] and PRT6[4,5]. Stability is also increased through this pathway in the absence of endogenous NO[17] or by perturbing mitochondrial function[18]. In addition, environmental reduction of either oxygen or NO, increased ethylene, low temperature, heavy metal stress and herbicides also enhance ERFVII stability[8,15,19–21]. ERFVII degradation is also controlled by other E3 ligases, in PCO N-degron pathway independent, mechanisms[22,23].

Both $O_2$ and NO-sensing through the PCO N-degron pathway are important for ERFVII (and other substrate) regulated tolerance to multiple abiotic stresses. Sensing of low oxygen enhances plant

tolerance of submergence and waterlogging, and is an important component of plant agricultural and natural environmental adaptation to acute hypoxia[24–28] as well as chronic hypoxia[9,29–33]. Sensing of NO through ERFVIIs contributes to tolerance to salinity, drought and heat[17] and has also been shown to pre-acclimate plants to low oxygen stress[19]. Oxygen-sensing through ERFVIIs has a role in environmental adaptation. The rice ERFVII SUBMERGENCE (SUB)1A, a key locus for improving rice submergence tolerance, was shown not to be a substrate of the PCO N-degron pathway, despite being a $Met^1$-$Cys^2$ initiating protein because the carboxyl terminal structure prevents its degradation[4,34]. Recently it was shown that adaptation to geographical aridity is associated with differences in alleles of two cis-elements upstream of the ERFVII *RELATED TO AP* (*RAP*)2.12 specifically within the species *Arabidopsis thaliana*[27], and within angiosperms alteration of the sensitivity of oxygen-sensing is associated with adaptation to altitude[26].

Importantly, although many components of the plant oxygen-sensing mechanism have been defined, several key aspects remain unresolved, un-investigated or inferred from previous work in mammalian systems. For example, it has not been shown that substrates are degraded through the proteasome via ubiquitylation, or the relative activity of the PCO N-degron pathway in relation to substrate degradation in response to hypoxia. Although in vitro analysis using recombinant PCOs and ERFVII oligopeptides showed oxidation of amino-terminal Cys to Cys-sulfinate $(SO_2^-)$[12], in vivo data from substrates purified from mouse L cells showed that the amino-terminal Cys of mammalian Regulator of G protein Signalling (RGS)4 was Arg-$Cys(SO_3^-)$ (sulfonate), not sulfinate[35]. More recently work in mammalian systems has suggested that $Cys(SO_3^-)$ may also be a product of oxidative stress, resulting in mammalian substrate degradation through p62-mediated autophagy[36]. Finally, although ERFVIIs have been shown to activate hypoxia-related gene expression, their combined effect on the regulation of genome expression is not known. Here we experimentally address important gaps in understanding of the known plant oxygen-sensing system, defining the role of ERFVIIs in regulating gene expression though the PCO N-degron pathway, their influence in root responses to acute hypoxia, their modulation through environmental and genetic factors, and the influence of oxygen on the chemical nature of the amino-terminus.

## Results

### ERFVIIs non-autonomously enhance tolerance to acute hypoxia

To analyse the influence of ERFVIIs on physiological responses to different forms of acute hypoxia we used a genetic approach removing the activity of all five *A. thaliana* ERFVIIs in the *rap2.12 rap2.2 rap2.3 hypoxia responsive erf* (*hre*)1 *hre2* pentuple mutant (hereafter abbreviated to *erfVII*) and *prt6 erfVII*[28]. Comparison of *prt6 erfVII* with the single E3 ligase mutant *prt6* allows an understanding of the effect of constitutive stabilisation of ERFVII proteins (in *prt6*) against a background without *ERFVII* activity (*prt6 erfVII*). Analysis of tolerance of seedling root tips to severe rapidly evoked hypoxia (<0.5% $O_2$) showed that *prt6* greatly enhanced survival, as was previously shown[19], but removal of all ERFVII activity in the *prt6* background (*prt6 erfVII*) strongly reduced tolerance (Fig. 1b, Supplementary Fig. 1a). In comparison, *erfVII* root tips showed a similar tolerance to wild-type (accession Col-0) seedlings but tolerance was reduced compared to *prt6 erfVII*. To determine the individual contribution of each *ERFVII* we analysed root tip survival of hypoxia in the *prt6* genetic background in which only one of the five ERFVIIs was functional (for example, *prt6 RAP2.12 rap2.2 rap2.3 hre1 hre2*). This analysis showed that no single ERFVII was capable of enhancing survival to the level of *prt6* (where all ERFVIIs are stable) (Fig. 1b). Conversely, each individual constitutively stabilised ERFVII (through conversion of $Cys^2$ to Ala [$C^2A$] derived from a *promoterERFVII:MA-ERFVII* transgene[28]) in a background of WT *ERFVII* activity could significantly increase root tip hypoxia survival compared to Col-0 untransformed control (Fig. 1b). $C^2A$-ERFVIIs could not

enhance hypoxia-related gene expression (with the exception of *SUS4* for MA-RAP2.12 (Supplementary Fig. 1c). Both RAP2.12 and RAP2.2 when individually stabilised by the *prt6* background did enhance expression of the three core hypoxia transcripts analysed, and RAP2.3 one transcript (Supplementary Fig. 1c). Hypoxia negatively affects seed germination and a role for ERFVIIs through oxygen-sensing was previously proposed[4]. We analysed the role of ERFVIIs in regulating seed germination under hypoxia (Fig. 1c, Supplementary Fig. 1b), that showed a key role specifically for *RAP2.2* in enhancing germination potential at low oxygen. The influence of *ERFVII*s on aerial and root biomass response to prolonged (7 days) waterlogging in natural soil was also investigated. Both dry and fresh weights of rosettes and roots of wild-type plants were reduced in response to treatment, however reduction of biomass was far greater in the *erfVII* mutant (Fig. 1d, e). To understand the effect of waterlogging on root development and 3D-architecture in situ in the soil we used non-invasive X-ray computed micro-tomography (μCT) imaging of wild-type and *erfVII* roots of 4-week-old plants grown in well-watered (control) or waterlogged mixed sandy-clay loam soil for 7 days (Fig. 2a, b, Supplementary Fig. 2, Supplementary Movies 1 and 2). μCT imaging of control well-watered soil showed no appreciable difference in root development, whereas in waterlogged soil root total volume, maximum depth and surface area (referred as region of interest, ROI) were significantly reduced both in wild-type and *erfVII* plants, with a more drastic reduction observed in the *erfVII* mutant. In particular, total root volume and surface area in waterlogged conditions decreased by 99 and 92% reduction respectively in *erfVII* compared to 29 and 26% reduction in Col-0.

### Defining the influence of ERFVIIs on the seedling transcriptome

To understand the influence of *ERFVII*s on global transcript abundance we compared transcriptomes using RNAseq data derived from 7-day-old seedlings grown in continuous light of Col-0 (wild-type), with *prt6* and *prt6 erfVII* (Supplementary Data 1–3). Principal Component Analysis (PCA) showed that in comparison to *prt6*, removal of *ERFVII* genetic activity in *prt6 erfVII* strongly influenced gene expression in the *prt6* background, suggesting that ERFVIIs promote a large effect on the transcriptome of *prt6-1* (Supplementary Fig. 3a). In *prt6-1* (where all substrates of the PRT6 N-degron pathway are stabilised) only ~2.5% of the genome was differentially regulated compared to Col-0, as opposed to 25% compared to *prt6 erfVII*, and 28% comparing Col-0 to *prt6 erfVII* (Supplementary Data 1–3). Interestingly comparison of *prt6* vs *prt6 erfVII* with Col-0 vs *prt6 erfVII* showed a very high level of similarity of differentially (both up and down)-regulated genes (Fig. 3a) suggesting that ERFVIIs are at least partially active in Col-0 under normoxia. Analysis of Gene Ontologies demonstrated that the major functional groups associated with *ERFVII* genetic activity in the comparison of *prt6* to *prt6 erfVII* were related to biological processes of response to hypoxia (Fig. 3b, Supplementary Data 4, 5). Analysis of genes previously shown through meta-analysis of available transcriptome datasets to be differentially regulated in response to hypoxia[37] showed that almost half were shared with transcripts differentially regulated in *prt6* compared to *prt6 erfVII* (Fig. 3c). The majority of a set of 49 core hypoxia-responsive transcripts, previously shown to be upregulated in diverse cell populations of *A. thaliana* seedlings[38] were also more highly expressed in *prt6* than *prt6 erfVII*, more highly in *prt6* than Col-0 and in hypoxia compared to normoxia[15] (Fig. 3d, Supplementary Data 6). Out of 49 core hypoxia genes 30 were differentially regulated in *prt6* compared *to prt6 erfVII* and also shown previously through genome wide chromatin immunoprecipitation (ChIP) to be bound by the ERFVII HRE2 under hypoxic stress (Supplementary Data 6)[39]. These included *PCO1* and *PCO2*, core components of the oxygen sensing pathway (Fig. 1a). Moreover, many genes differentially expressed between *prt6* and *prt6 erfVII* contain copies of the Hypoxia Responsive Promoter Element (HRPE)[40] in the −1000bp promoter region or genomic locus region (Supplementary Data 7–10).

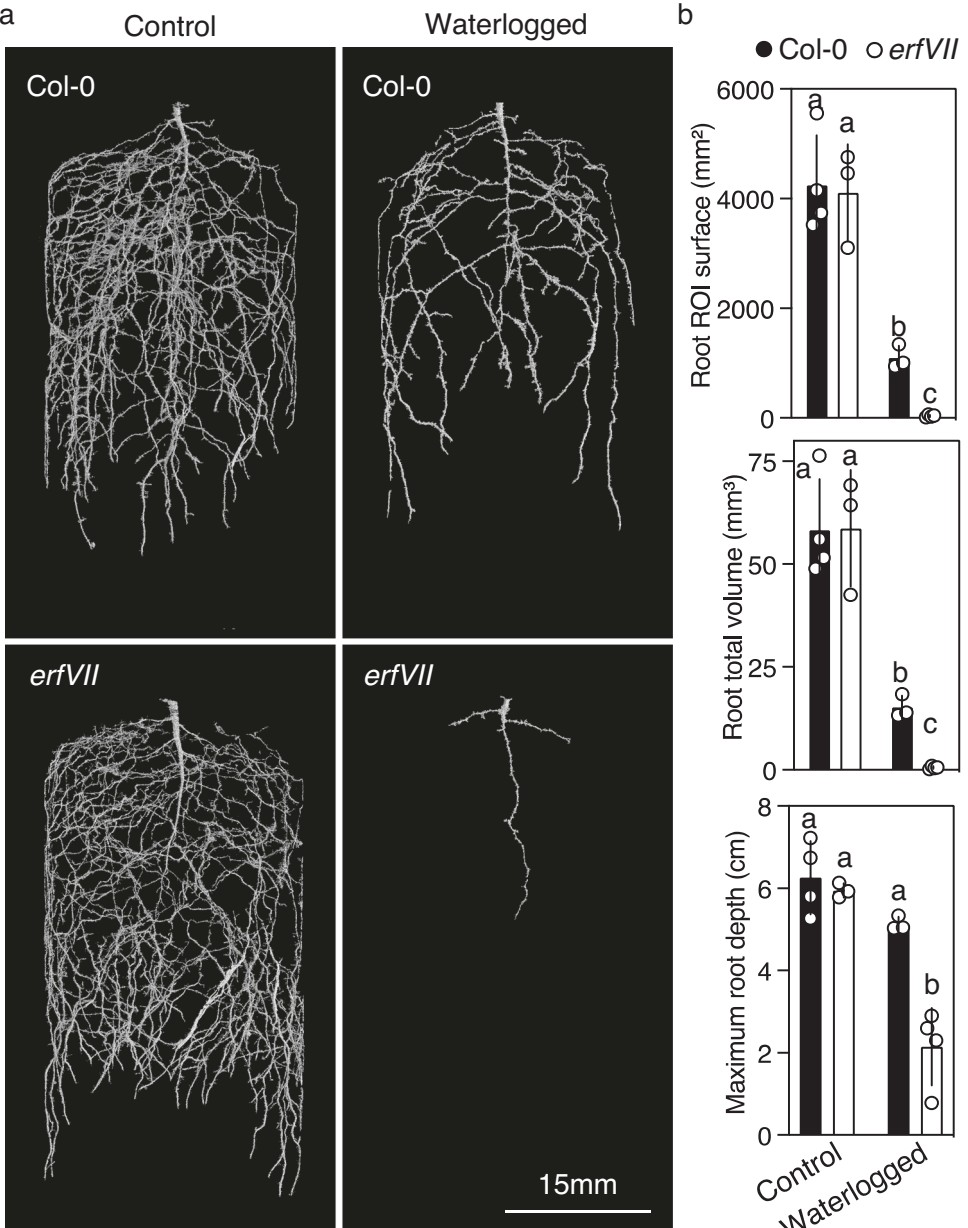

**Fig. 2 | ERFVII influence on root structure in soil. a** Representative 3D rendered X-ray computed micro-tomography front view images of four-week-old wild-type (Col-0) and *erfVII* mutant roots grown in sandy-clay loam soil under control conditions or waterlogging treatment conditions for 7 days. Scale bar is 15 mm. **b** Root total volume (mm³), root region of interest (ROI) surface area (mm²) and maximum root depth calculated from X-ray computed micro-tomography images of wild type (Col-0) and *erfVII* mutant roots with or without waterlogging treatment, as described in (**a**). Data are presented as mean ± SD (*n* = 4). For μCT imaging, four replicates were imaged, data were analysed by two-way ANOVA and different letters indicate significant differences (*p*-value < 0.05). Scale bar is 15 mm.

Genes upregulated in *prt6* that contain the HRPE and were shown to bind HRE2 in vivo showed stronger over-representation of GO terms associated with "hypoxia" than those upregulated in *prt6 erfVII* (Supplementary Fig. 3B, Supplementary Data 11).

### Genetic and environmental determinants controlling ERFVII stability

Although several genetic and physiological components have been shown to increase stability of ERFVII proteins no direct comparison of the relative influence of these components has been carried out. Here we analysed how ERFVII protein steady-state level is influenced by genetic and environmental factors, using as an exemplar RAP2.3. Using a *35S:RAP2.3*[15] transgene construct[15], immunolocalization showed that RAP2.3[3xHA] was undetectable in control 4-day-old seedling roots, but was strongly stabilised and localised in the

nucleus after 3 h treatment with the proteasome inhibitor bortezomib (Fig. 4a). Using a ubiquitin fusion technique (UFT[41,42]) construct previously designed for analysis of protein stability in animal systems[43], we developed a plant transformation vector that provides expression of a single protein cleaved by constitutive deubiquitinase activity to reveal C-terminal Cys²-RAP2.3[3xHA] (Supplementary Fig. 4a). This construct has the advantage of monitoring RAP2.3 stability without the requirement for MetAP activity to remove the initiating Met[1]. In wild-type Col-0 accession seedlings grown under constant light RAP2.3[3xHA] abundance was very low, but was significantly enhanced in the presence of the proteasome inhibitor bortezomib, where clear addition of high molecular weight poly-ubiquitin moieties was visible (Fig. 4b, Supplementary Fig. 4b). In comparison in the *prt6-1* background RAP2.3[3xHA] was also stabilised, but not to the same degree as in the presence of bortezomib.

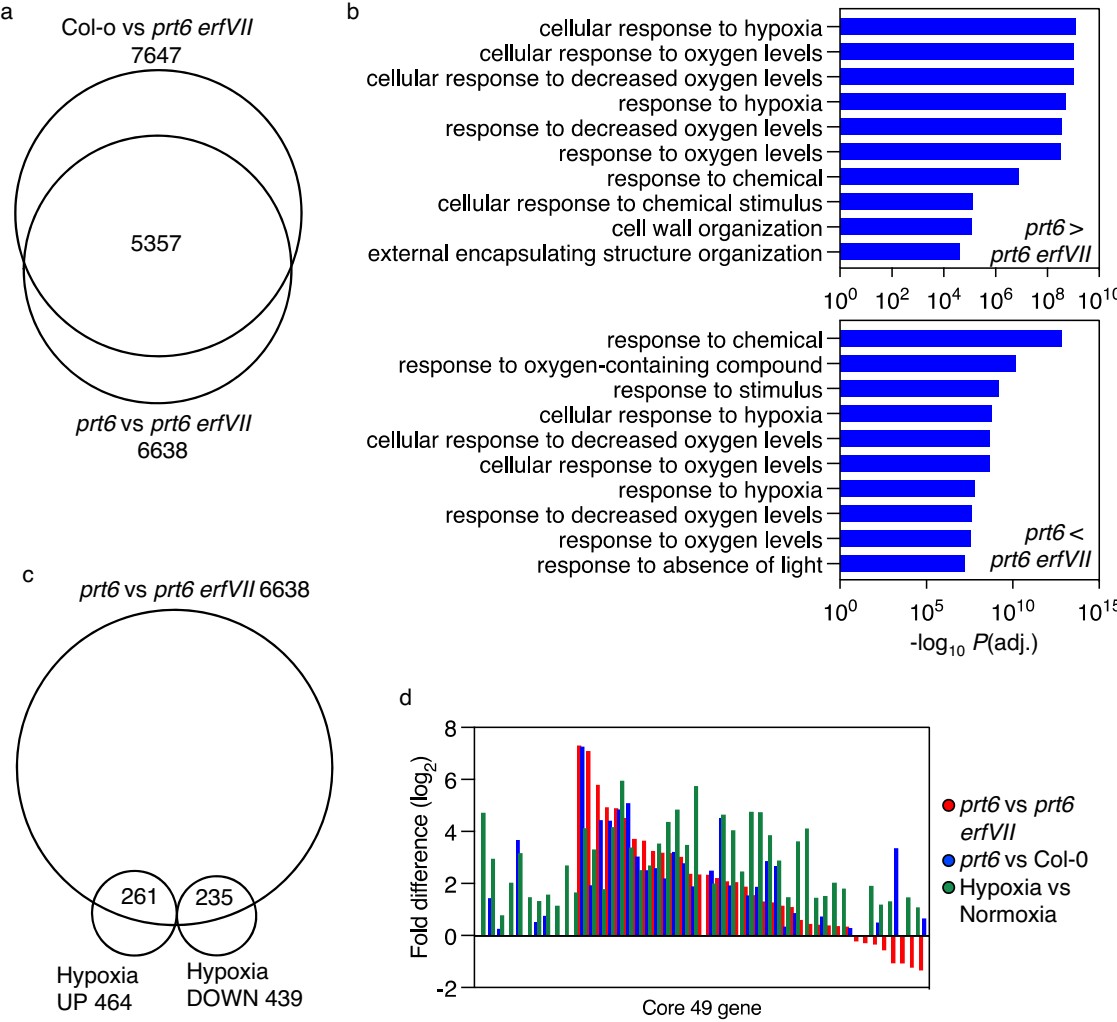

**Fig. 3 | Influence of *ERFVII*s on the seedling transcriptome. a** Proportional Venn diagram representation of gene sets for Col-0 (wild-type) vs. *prt6 erfVII* and *prt6* vs. *prt6 erfVII*. In each case numbers of total differentially regulated genes (up or down) are shown. **b** Top 10 most significant GO terms associated with transcripts differentially regulated in *prt6* vs. *prt6 erfVII*, Fisher's one-tailed test (*p* < 0.05). **c** Proportional Venn diagram representation showing overlapping gene sets between genes differentially expressed between *prt6* vs *prt6 erfVII* and hypoxia vs normoxia[37]. Total number of differentially regulated genes are shown next to comparisons, in set identities within the intersecting set. **d** Differential expression of a set of core 49 hypoxia upregulated genes[38] in transcriptome comparisons, sorted by fold difference in *prt6* vs *prt6 erfVII*.

Addition of bortezomib increased stability in *prt6-1* and also resulted in a polyubiquitin ladder (Fig. 4b).

The influence of two forms of reduced oxygen availability on RAP2.3[3xHA] stability were assayed, either by submergence in the dark or dark incubation in a chamber with reduced oxygen (hypoxia). Whereas the former results in entrapped intracellular ethylene the latter does not. RAP2.3[3xHA] protein was rapidly stabilised by both treatments in comparison to control seedlings left in the dark, for submergence that occurred within 10 minutes (Fig.4c, Supplementary Fig. 4c). ERFVII-regulated *ADH1* (Supplementary Data 3) was rapidly induced under both conditions after the stabilisation of RAP2.3[3xHA] (Fig. 4d), as were ERFVII-regulated *PCO1* and *SUS4* (Supplementary Fig. 4c). To determine the influence of MetAP activity (Fig. 1a) we analysed protein accumulation in submergence for the *35 S:RAP2.3*[3xHA] transgenic line used for immunolocalization (Fig. 4a). Stabilisation of RAP2.3[3xHA] in this line showed very similar kinetics to RAP2.3[3xHA] from the UFT transgene, indicating that MetAP activity does not influence accumulation kinetics (Supplementary Fig. 4e). Following return to normoxia, RAP2.3[3xHA] abundance reduced to very low levels within 30 to 60 min (Fig. 4e). Previously it was shown that PCO enzymes require $Fe^{2+}$ to oxidise amino-terminal $Cys^{10}$ and zinc deficiency was shown to enhance ERFVII

stability[21]. Constant growth of seedlings or transfer to media +Fe or -Fe showed that under $Fe^{2+}$ starvation conditions RAP2.3[3xHA] was stabilised (Fig. 4f). In comparison to bortezomib, RAP2.3[3xHA] abundance in response to physiological treatments that reduce oxygen or NO was lower, indicating either incomplete inactivation of the PCO N-degron pathway or alterative mechanisms for degrading ERFVIIs (Fig. 4g). Abundance was higher in the presence of the NO scavenger cPTIO than under reduced oxygen conditions.

Finally, we analysed the influence of genetic adaptation to altitude in the regulation of ERFVII stability in relation to atmospheric oxygen. We analysed two accessions of *A. thaliana* originally collected at 100 m above sea level (Col-0 $pO_2$ 21 kPa) or 3400 m (Sha $pO_2$ 13.9 kPa). In etiolated seedlings at $pO_2$ 21 kPa RAP2.3[3xHA] was more abundant in Col-0 than Sha. However, under lower ambient oxygen (15%, equivalent to 14.7 kPa) RAP2.3[3xHA] was as abundant in high altitude accession Sha as in low altitude Col-0 (Fig. 4h).

## Characterisation of ERFVII amino terminal modifications

We investigated the chemical changes that occur in vivo at the amino-terminus of ERFVIIs as a result of the action of the PCO N-degron pathway. To do this we employed a wheat germ extract commonly

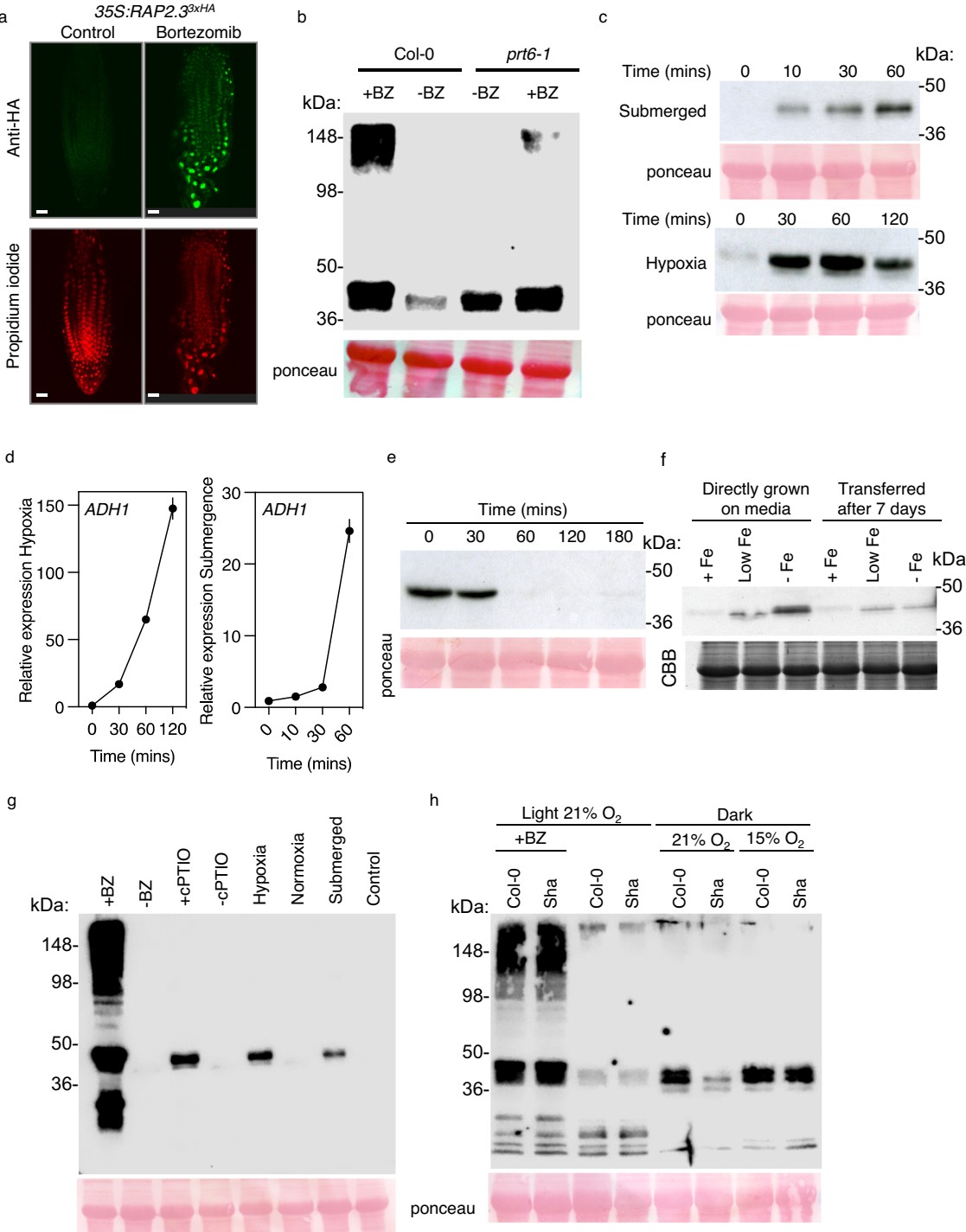

**Fig. 4 | Modulation of RAP2.3 stability by genetic and environmental factors.**
**a** In situ immunolocalization of RAP2.3$^{3xHA}$ in control or bortezomib (50 μM) treated roots of 4-day-old seedlings, scale bars 20 μm. **b** Western blot analysis of RAP2.3$^{3xHA}$ abundance in control or bortezomib treated WT or *prt6-1* mutant seedlings. **c** Time-course Western blot analysis of RAP2.3$^{3xHA}$ abundance in WT seedlings in response to submergence or hypoxia treatment. **d** Time-course of expression of *ADH1* in WT seedlings in response to submergence or hypoxia treatment. Data are presented as mean ± SD (*n* = 3). **f** Western blot analysis of RAP2.3$^{3xHA}$ abundance in WT seedlings in response to constant contact or transfer to media with no, low or high supplemented Fe$^{2+}$. **e** Western blot analysis of RAP2.3$^{3xHA}$ abundance in WT seedlings following transfer from hypoxia (60 min) to normoxia. **g** Western blot analysis showing relative abundance of RAP2.3$^{3xHA}$ in response to bortezomib (BZ), cPTIO (NO scavenger), hypoxia or submergence. **h** Western blot analysis showing relative abundance of RAP2.3$^{3xHA}$ in low altitude accession Col-0 in comparison to high altitude Sha at either 21% or 15% ambient oxygen. BZ treatment for light-grown seedlings. Ponceau or CBB staining of Western blots is shown.

used for in vitro coupled transcription/translation. Unlike the rabbit reticulocyte lysate (RRL), this system was previously shown not to contain an active UPS[44]. Therefore, whereas Cys2 proteins are synthesised and degraded in the RRL through the oxygen-dependent action of the Arg/N-degron pathway[6,7,45], this does not occur in the wheat germ extract. We used this plant-derived extract to synthesise ERFVIIs and study chemical changes at the amino terminus. Following coupled transcription/translation in wheat-germ lysate and anti-HA-based pull-down of RAP2.3$^{3xHA}$ and RAP2.12$^{3xHA}$ bands were visible on Coomassie-stained SDS-PAGE gels at the expected sizes of the two

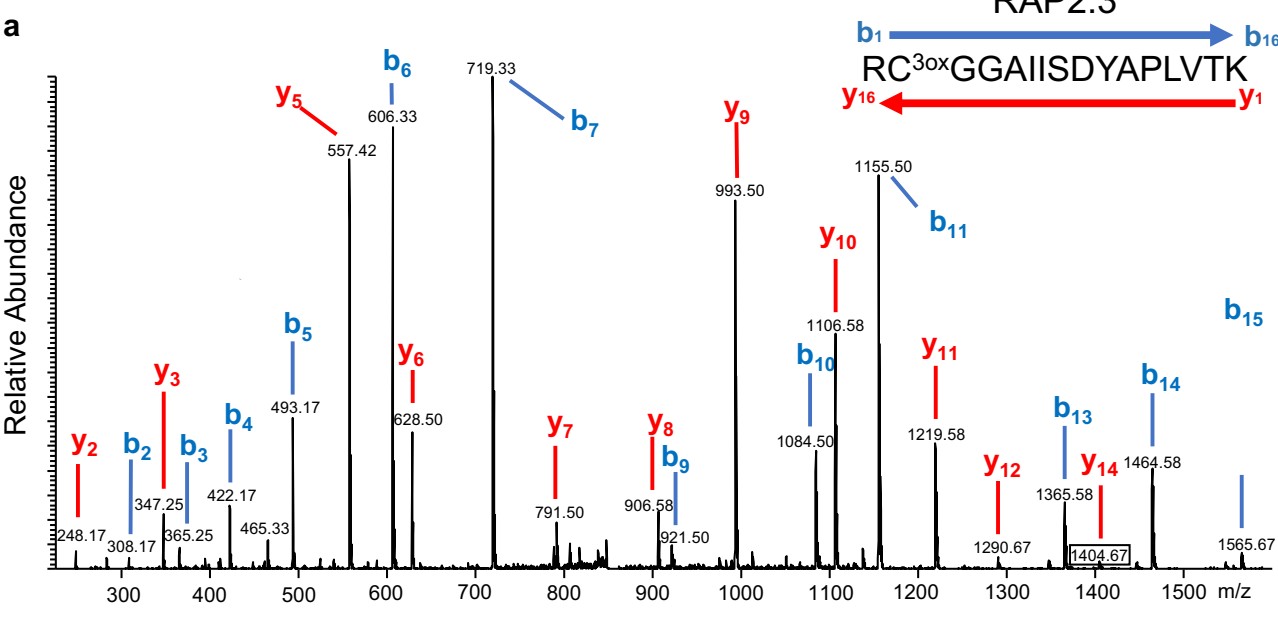

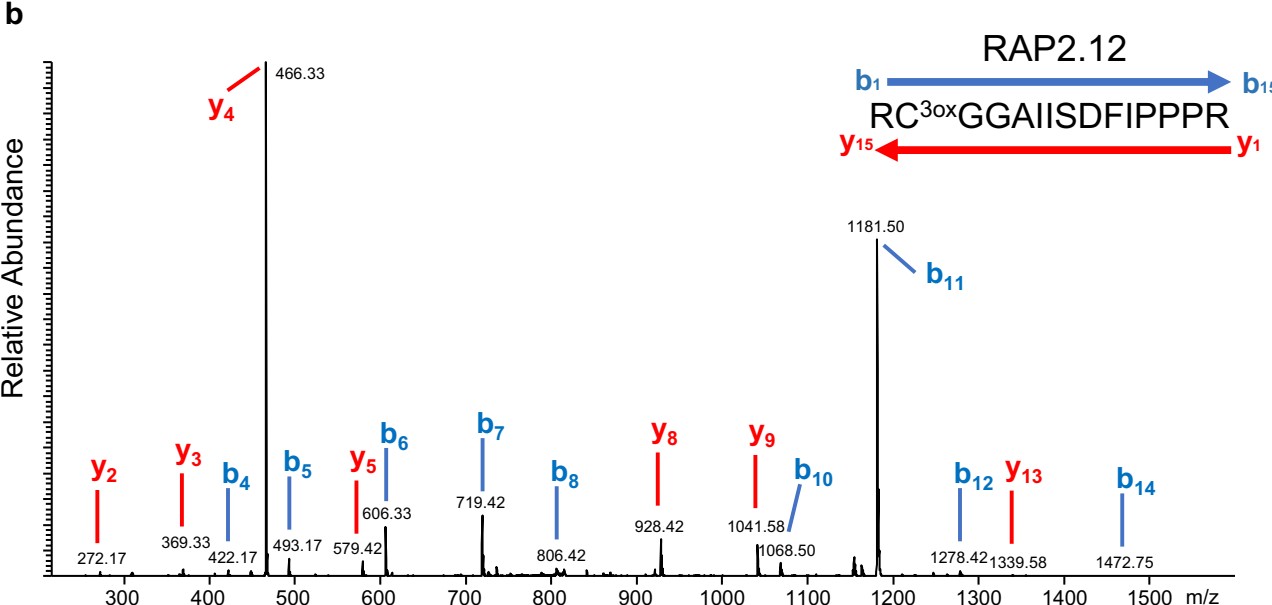

**Fig. 5 | Analysis of the chemical structure at the amino-termini of RAP2.3 and RAP2.12 in a plant extract.** MS/MS spectra of the Nt tryptic peptides of (**a**) RAP2.3 ([M + 2H]2 + = $m/z$ 856.4) and (**b**) RAP2.12 ([M + 2H]2 + = $m/z$ 823.9) following expression of RAP2.3$^{3xHA}$ and RAP2.12$^{3xHA}$ proteins, respectively, in a wheat germ extract showing the presence of Cys-sulfonic acid (C$^{3ox}$) and arginylation at the amino-terminus (R-) in each case.

proteins (Supplementary Fig. 5a). Excision, destaining and trypsin digestion of these bands, followed by LC-MS/MS analysis and database searching, identified them unambiguously as originating from RAP2.3 and RAP2.12 12 based on the presence of 9 matching peptides from each protein (Supplementary Fig. 5b, Supplementary Data 12). Examination of the MS and MS/MS data revealed the same extensive alterations at the amino termini of both proteins, including methionine excision, and arginylation. In addition, a Cys-sulfonate (SO$_3^-$) modification only at position 2 was also observed. High quality MS/MS spectra containing a clear series of b and y fragment ions were recorded for the two tryptic peptides to confirm this observation (Fig. 5a, b). Interestingly, no signal for Nt-Arg-Cys-sulfinate (SO$_2^-$) was detected. As a negative control for artefactual Cys oxidation, RAP2.3$^{HIS}$ was synthesised in *E. coli*, which does not possess the enzymatic machinery for Nt-Cys oxidative processing. As expected, no oxidation

or subsequent arginylation was detected, and the unmodified Nt-tryptic peptide was found using the LC-MS/MS methodology with the Met[1] removed (from the action of endogenous *E. coli* MetAP activity; Supplementary Fig. 5c).

Previously it was shown that the two oxygens added to amino-terminal Cys by PCOs originate from molecular oxygen not water[12]. To establish whether the third oxygen seen in the Cys-sulfonic acid residue of RAP2.3 originated from water or molecular oxygen, RAP2.3$^{3xHA}$ was synthesised using the wheat germ lysate in 40% (v/v) H$_2$$^{18}$O. If the additional oxygen originated from water, then a pair of Nt-peptide isotopologues bearing $^{16}$O:$^{18}$O in the ratio 60:40 would be expected. LC-MS analysis of the tryptic digest of RAP2.3$^{3xHA}$ synthesised under these conditions showed that $^{18}$O from H$_2$$^{18}$O was incorporated into the RAP2.3 peptide (Supplementary Fig. 5d). MS/MS performed on the [M + 2H]$^{2+}$ precursor ions at $m/z$ 856.5/857.5 showed that $^{18}$O was

actually incorporated into the carboxylate group of the Asp-9 residue and that no $^{18}O$ was incorporated into Cys2 (Supplementary Fig. 5e). We infer from this result that the third oxygen atom seen in the Cys sulfonate residues of both RAP2.3 and RAP2.12 is derived from molecular oxygen.

## Discussion

The ERFVII transcription factors are key components of the plant oxygen sensing response, regulating the expression of genes associated with tolerance to low oxygen[1,2]. In order to understand the influence on the genome of the entire *ERFVII* cohort we developed the pentuple *erfVII* mutant that removes all five *ERFVII* genetic functions. Combination with the *prt6* E3 ligase mutation (*prt6 erfVII*) allows an analysis of the specific role of *ERFVII*s in a genetic background where all other PRT6 substrates are stable (including ZPR2, VRN2[8,9] and also other as yet unknown substrates[16]). These two genetic resources, in combination with lower order permutations for *ERFVII* mutations allowed an analysis of their role(s) in regulating response to low oxygen at increasing physiological levels of complexity. Enhanced root tip tolerance to acute hypoxia of the *prt6* mutant was shown to require *ERFVII* function, as tolerance was strongly reduced in *prt6 erfVII*, similarly previously described for lower order combinations of *erfVII* mutations[19]. Interestingly removal of *ERFVII* function in the *prt6* background does not reduce tolerance to the same level as *erfVII*, suggesting that other PRT6 substrates contribute to this response. In a *prt6* background with all other *ERFVII*s genetically removed each individual WT *ERFVII* was unable to enhance hypoxia tolerance above the level shown by *prt6 erfVII*. However, individually stabilised ERFVII proteins (through transgenes containing $C^2A$) in a genetic background with WT endogenous *ERFVII*s were able to strongly enhance root tip survival, all to a similar level. This indicates that this sub-family functions in a non-autonomous fashion, each ERFVII requiring the activity of others to enhance root tip hypoxia survival. Conversely hypoxia gene expression in normoxia was more greatly affected by individual stabilised ERFVIIs in the *prt6* background than by individual $C^2A$-ERFVIIs in a wild-type background, that may indicate that perception of hypoxia itself is also important for ERFVII action in enhancing tolerance. Although *RAP2.12* has been shown to be the predominant ERFVII regulating response to submergence in *A. thaliana* accessions adapted to humid environments (including Col-0)[27], germination tolerance to low oxygen was shown to be dependent on active *RAP2.2*. This suggests some functional diversification of ERFVIIs in the hierarchy of regulation of low oxygen responses at different stages of plant development, where different responses to hypoxia may be required.

The most important interface between plants and water availability are roots, that also face the challenge of growth under the soil in potentially hypoxic conditions when water levels rise due to flooding[46,47]. The role of oxygen-sensing in regulating lateral root formation was previously investigated in plants grown on agar medium, where it was shown that *ERFVII*s repress auxin signalling thereby inhibiting lateral root production[30]. Bending of primary roots in response to external hypoxia was also reported to be dependent on ERFVIIs and alteration of auxin signalling, again in plants grown on agar medium[48]. However, no analysis has previously been undertaken to understand how ERFVIIs influence root 3D development in natural soils. To understand the role of *ERFVII*s in roots growing in soil we used non-invasive 3D X-ray μCT imaging. This allowed an analysis of the tridimensional root architecture under non-stressed and waterlogged conditions, highlighting the strong reduction of root development (more than 90%) that occurs in natural soil in the absence of *ERFVII*s (Fig. 2a, b, Supplementary Fig. 2, Supplementary Movie 1, 2). Analysis of aerial rosette growth showed a very strong response to root waterlogging, including reduced growth and increased anthocyanin synthesis, once again with a more severe effect in the *erfVII* mutant

compared to wild-type. A slightly reduced growth, although not significant, was also observed in *erfVII* roots and rosettes under non-stressed conditions, suggesting a role of *ERFVII*s in plant growth and development, worthy of further investigation. This observation is in agreement with the recent finding that lack of functional *ADH1* or *PDC1* leads to a growth penalty under normal (aerobic) growth conditions[49]. These results highlight the importance of ERFVIIs for root function in the soil and suggests that evolution of *ERFVII*s was an important determinant in the ability of roots to survive in hypoxic soil environments.

Having shown the strong effect of *ERFVII*s on physiological responses to low oxygen, we analysed total *ERFVII*-regulated gene expression by comparing *prt6* with *prt6 erfVII*. Analysis of differentially expressed genes highlighted the importance of *ERFVII*s for regulating the expression (both positively and negatively) of transcripts involved in response to low oxygen (Fig. 3, S3 and Supplementary Data 4, 5). *ERFVII*s were shown to regulate the expression of around 25% of the genome in seedlings. In addition, ten times the number of genes were differentially expressed in comparisons including *prt6 erfVII*, compared to Col-0 vs. *prt6*, indicating the profound influence of stabilised ERFVIIs on gene expression. Interestingly the transcriptomic state of Col-0 is much more similar to *prt6* than *prt6 erfVII*, a feature highlighted by analysis that showed a strong overlap of similarly differentially expressed genes (Fig. 3a). This suggests that in Col-0 seedlings grown in continuous light ERFVIIs are at least partially active (and therefore stable), as was recently observed for dark grown etiolated seedlings[26] and suggested by the repression of oxygen-sensing by mitochondrial retrograde signalling during normal development[18]. Previous work has highlighted the importance of environmental stress in early growth stages of plant development[50,51], suggesting that some level of stabilisation of ERFVIIs during seedling establishment may have adaptive advantage. In line with this suggestion, it has also been shown that ERFVIIs are more active at early growth stages compared to later development in *A. thaliana*[29,52]. Approximately 50% of hypoxia-associated transcripts were differentially expressed in *prt6* compared to *prt6 erfVII*, suggesting that ERFVIIs may not control the complete response to low oxygen, and that other mechanisms contribute to the hypoxia-related transcriptome[53–56]. The HRPE, previously identified in the LBD41 promoter[40], and shown to be a key cis-element associated with ERFVII function[39,40] was present in the promoter or gene regions of many genes whose transcripts are differentially regulated by *ERFVII*s, suggesting direct regulation of these genes. In addition, there was a strong over-representation of hypoxia-related transcripts in the overlapping set representing genes differentially regulated by *ERFVII*s that contain the HRPE and that also were shown to bind HRE2 in genome-wide ChIP analysis[39]. In combination with previous datasets analysing the role of individual or combinations of ERFVIIs[19,40,57], these data highlight the overall importance of *ERFVII*s in regulating the expression of the genome in response to low oxygen, as a response to inhibition of the PCO N-degron pathway.

Direct visualisation of ERFVII abundance through immunolocalization and Western blot analysis allowed an investigation of the relative influences of environmental and genetic factors. Stabilisation of RAP2.3$^{3xHA}$ in the presence of bortezomib and addition of polyubiquitin moieties demonstrate that degradation of ERFVIIs requires the proteasome (Fig. 4a, b, Supplementary Fig. 4). Genetic removal of PRT6 activity also resulted in stabilisation, though not as strongly as with bortezomib, and addition of bortezomib to *prt6* seedlings revealed high molecular weight products, suggesting the activity of other E3 ligases in regulating ERFVII stability. In addition to PRT6, two other *A. thaliana* proteins carry a UBR-like motif (required for Arg N-degron substrate recognition), BIG and the product of AT4G23860[16] and other E3 ligases unrelated to the N-degron pathway also regulate ERFVII stability[22]. Not unexpectedly, because PCO activity requires $Fe^{2+}$, growth on low iron medium resulted in increased RAP2.3$^{3xHA}$

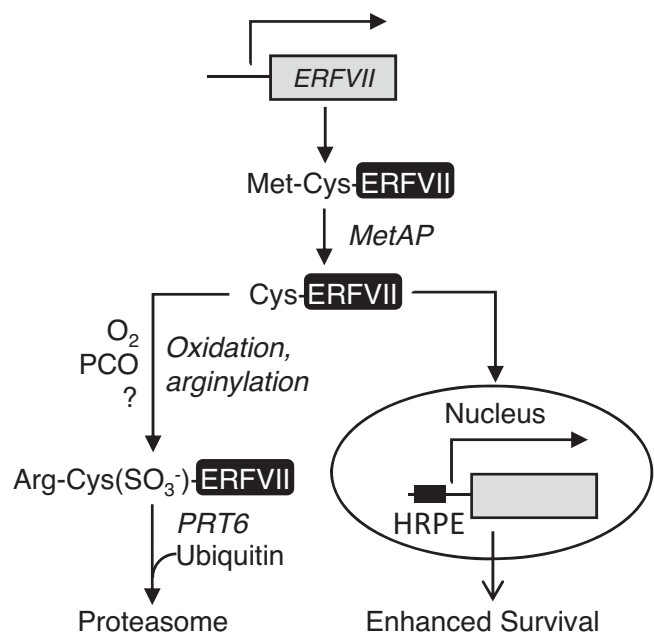

**Fig. 6 | Model for plant oxygen sensing through the PCO N-degron pathway based on data presented in this work.** *ERFVII* transcription is shown at the top of the model. Subsequent enzymatic steps on the protein are shown, either oxygen-independent (MetAP) or oxygen dependent. Activation of nuclear gene expression by stabilised ERFVIIs in the absence of oxygen, through the Hypoxia Responsive Promoter Element (HRPE) is indicated. "?" indicates an unknown component required for the addition of a third oxygen to form Cys sulfonate.

abundance, suggesting that ERFVII function might be an important component of the plant low iron response as shown for the response to zinc excess[21]. Both hypoxia treatment and submergence resulted in rapid stabilisation of RAP2.3[3xHA], for submergence within 10 minutes, preceding accumulation of RNA for ERFVII-regulated transcripts. Such a rapid onset of stabilisation indicates that under dark submergence intracellular oxygen is used up very rapidly resulting in strong inhibition of PCO activity. A model for the function of RAP2.12 has been proposed whereby under normoxia a population of molecules is tethered to the plasma membrane and released under hypoxia to migrate to the nucleus and activate gene expression[5]. Whilst not precluding this possibility, our results indicate that a simpler explanation only requires that ERFVIIs are stable after translation following very rapid intracellular oxygen depletion and subsequently translocate to the nucleus and activate transcription (Fig. 6). RAP2.12 has been studied as the most influential ERFVII in *A. thaliana*, a recent study showed that this is because a subset of accessions of this species, including Col-0, are adapted to humid environments through specific cis-element changes that enhances constitutive expression of *RAP2.12*[27]. This indicates that in other species RAP2.12 orthologues may not necessarily take a predominant role. We also observed that none of the five ERFVIIs could autonomously enhance root hypoxia tolerance, that required a combination of ERFVII action (Fig. 1b).

Previous work showed that recombinant PCOs can oxidise amino-terminal Cys to Cys-sulfinate (SO₂)⁻ that can act as a substrate for ATE[10,12], but when purified from mouse L cells the amino-terminus of Cys[2] protein RGS4 was shown to be in the form Arg-Cys(SO₃)⁻[35], that was also shown to be the case in response to oxidative treatments of different cell types[36]. Using a wheat-germ extract we observed the formation of Cys(SO₃)⁻ and arginylation of both RAP2.3 and RAP2.12. Analysis in the wheat germ extract system precludes oxidation through ROS or other chemical oxidising agents as synthesis is carried out in the presence of DTT. PCOs are dioxygenases and thus catalyse the incorporation of two oxygen atoms from molecular oxygen into the amino-terminal Cys residue. Although it is known that this degree of oxidation is sufficient for recognition by ATE1, and hence arginylation of amino-terminal Cys proteins[12], in our analyses using wheat germ extract we were only able to detect arginylated peptides possessing the sulfonate (SO₃)⁻ level of oxidation. This raises the question of the origin and significance of the third oxygen atom in the sulfur acid. As the third oxygen was derived from molecular oxygen, we suggest that it is most likely introduced by the action of a yet uncharacterised enzyme. This monooxygenase/oxidase presumably acts after PCO oxidation, but whether it precedes arginylation or not remains an open question given that ATE1 is able to act on proteins bearing either amino-terminal Cys sulfinic or sulfonic acids[12]. As the pathway is highly conserved in eukaryotes[2] (with ADO acting as the oxygen sensing enzyme in mammals[14]) it is possible that such an enzyme also exists in non-plant systems.

Here we investigated the influence of ERFVIIs on outputs of the PCO N-degron oxygen-sensing pathway. Our observations provide a holistic understanding of the action of these transcription factors in influencing genome expression through PRT6 gated stability, and on root physiological response to low oxygen at increasing environmental complexity and on how their activity is modulated. In particular, we provide evidence of the importance of *ERFVII*s in maintaining root growth in hypoxic soil conditions associated with waterlogging. We defined exact chemical changes at the amino-terminus of ERFVIIs suggesting components that are required for oxygen-sensing through this pathway in both plant and animal systems. These data indicate avenues for future research to completely define oxygen-sensing through the PCO N-degron pathway and biotechnological approaches to enhance tolerance of crops to flooding and waterlogging.

## Methods
### Plant genetic resources
Several *A. thaliana* genetic resources used in this investigation were reported previously including *prt6-1* and *ate1 ate2*[58], *erfVII* and *prt6 erfVII*[28]. In all cases wild-type refers to the Col-0 accession of *A. thaliana*. Plants containing various combinations of *ERFVII*s with T-DNA insertions were obtained in the process of generating the *prt6 erfVII* sextuple mutant and contain the same alleles as this mutant. Accession Shakdara (Sha; NASC ID: N929, originally collected at 3400 metres above sea level at Pamiro-Alay: Tajikistan) containing pUFT:RAP2.3[3HA] was obtained by direct transformation of the accession[59].

### Analysis of plant growth and development
Plants were grown and seeds collected as described[60]. Analysis of root tip survival following hypoxia treatment (<0.5% ambient oxygen) and waterlogging were carried out as previous described[8,19]. Trypan blue staining was performed on roots to investigate cell death; seedlings were incubated with 0.004% Trypan blue solution for 10 mins then decolorized for 10 minutes in distilled water. Images were taken using the ZEISS Axio Observer microscope. The root tip map was imaged using a 20× objective under the brightfield function and fluorescence of trypan blue visualized by excitation of the samples with a 633 nm laser as described[61]. For soil-based root growth assay, wild-type and *erfVII* mutant seeds were germinated directly on soil, after three days of cold-treatment at 4 °C in the dark. Plants were grown for three weeks, followed by one week of waterlogging treatment, in controlled environmental conditions with photoperiod of 12:12 h light:dark. Hypoxia and submergence treatments for Western blot analysis of RAP2.3[3HA] abundance were carried out as previously described[8]. Seed germination under hypoxia was performed as previously described[4].

### Iron treatment
Iron treatment was done as described previously[62,63] using the *35S:RAP2.3*[3xHA] transgene. +Fe and −Fe plates were made with macro-nutrients and micronutrients at 2 mM Ca(NO₃)₂, 0.75 mM K₂SO₄,

0.65 mM $MgSO_4$, 0.1 mM $KH_2PO_4$, 10 µM $H_3BO_3$, 0.1 µM $MnSO_4$, 0.05 µM $CuSO_4$, 0.05 µM $ZnSO_4$, 0.005 µM $(NH_3)6Mo_7O_{24}$, 1 mM MES, 0.6% agar, adjusted to pH 6.0 and supplemented with 50 µM Fe(III)-EDTA for +Fe plates and ferrozine, an iron chelator [3-(2-pyridyl)−5,6-diphenyl-1,2,4-triazine sulfonate for low Fe (150 µM) and −Fe (300 µM) media.

## Soil material and X-ray µCT imaging

Sandy-clay loam soil was collected from the University of Nottingham farm at Bunny, Nottinghamshire, UK (52.52°N, 1.07°W), air dried and then passed through a 2 mm sieve to homogenise, as previously described[64]. Polyvinylchloride (PVC) columns (8 cm length × 3 cm diameter) were filled with the soil to a bulk density of ~1.1–1.2 g cm⁻¹. Four-week-old roots from control and waterlogged plants were imaged using a Phoenix v|tome|x M 240 high resolution X-ray µCT system (Waygate Technologies (a Baker Hughes business), Wunstorf, Germany) at the Hounsfield Facility, University of Nottingham, UK. The scanning parameters were optimized to allow balance between a large field of view and a high resolution. Prior to µCT scanning the soil columns were placed on a layer of drying paper for 2 days, to reduce soil moisture and thus enhancing contrast between root and the soil matrix. Each sample was then imaged using a voltage and current of 112 kV and 168 µA, respectively, at a voxel size resolution of 23 µm. The scanner stage rotated through 360 degrees at a rotation step increment of 0.125 degrees over a period of 48 minutes, acquiring a total of 2880 projection images using a fast (no image averaging) multiscan of three sections with an exposure time of 333 ms. Each scan was reconstructed using DatosRec software (Waygate Technologies (a Baker Hughes business), Wunstorf, Germany). Radiographs were visually assessed for sample movement before being reconstructed in 32-bit depth volumes with a beam hardening correction of 8, no filtration was applied to the image. Reconstructed sections were processed in VGStudioMAX (version 2.2.0; Volume Graphics GmbH, Heidelberg, Germany) with manual alignment of the 3 sections to combine to a single 3D volume.

## Image analysis

Merged volumes were processed in VGStudioMAX (version 2.2.0; Volume Graphics GmbH, Heidelberg, Germany) to identify plant root material. This was done via a 3D region growing algorithm to extract root architecture, as previously described in ref. 65. Once the segmented ROI was created, VGStudioMax was used to assess the rooting characteristics between treatments. The morphological properties measured were root volume ($mm^3$), root surface area ($mm^2$) and maximum in-situ rooting depth (cm).

## Affinity purification of RAP2.3³ˣᴴᴬ

For affinity purification of RAP2.3³ˣᴴᴬ, seven-day old seedlings grown in continuous light were treated for 2 h with 50 µM bortezomib (dissolved in DMSO). Total protein was extracted by grinding 0.5 g tissue in liquid nitrogen with 700 µl of IP buffer (50 mM Tris-Cl pH 8.0, 150 mM NaCl, 0.05% IGEPAL, 50 µM BZ, 100 mM PMSF). Extract was centrifuged at 13,000 rpm for 30 mins and supernatant (600 µl + 200 µl TBST) was incubated with 25 µl of Pierce™ Anti-HA Magnetic Beads (Thermo Fisher, UK) for 30 mins at room temperature. After incubation flow through was collected and beads were washed three times for 2 mins each with 300 µl of PBST (Tris Buffered Saline Pack (25 mM Tris, 0.15 M NaCl; pH 7.2, Thermo Fisher, UK; 0.1% (v/v) Tween-20). Finally, beads were washed with 300 µl molecular grade water. Elution of the beads was achieved by adding 100 µl of SDS-PAGE Lane Marker Non-Reducing Sample Buffer (2X), (0.3 M Tris-HCl, pH 6.8, 5% (v/v) SDS) (Thermo Fisher, UK) to the tube containing beads, followed by incubation at 95 °C for 10 min. Two µl of β-mercaptoethanol was added before loading onto an SDS-PAGE gel. Western blot analysis of RAP2.3³ˣᴴᴬ abundance were carried out as previously described[8]. Anti-

ubiquitin (Agrisera AS08 307 1:4000 dilution) and anti-HA antibodies (Sigma, H3663-200UL; 1:10,000 dilution) were used for Western immunodetection.

## Seedling transcriptome analyses

For RNA-seq analysis, seedlings of Col-0, *prt6-1* and *prt6 erfVII* were grown under continuous light on 1/2 strength MS agar. Four biological replicates of 7-day-old seedlings of each genotype were harvested and frozen in liquid nitrogen. RNA was extracted using RNeasy plant mini kit Qiagen as per manufacturer's instructions. RNA was sent to Deep-Seq Next Generation Sequencing Facility (University of Nottingham) for further analysis. RNA concentrations were measured using the Qubit Fluorometer and the Qubit RNA BR Assay Kit (ThermoFisher Scientific; Q10211) and RNA integrity was assessed using the Agilent 4200 TapeStation and the Agilent RNA Screentape (Agilent; 5067-5576). Poly(A) selection was performed on 1 µg of total RNA, using the NEBNext Poly(A) mRNA Magnetic Isolation Module (NEB; E7490L). Indexed sequencing libraries were then prepared using the NEBNext Ultra Directional RNA Library Preparation Kit for Illumina (NEB; E7420) and NEBNext Multiplex Oligos for Illumina, Index Primer Sets 2, 3 and 4 (NEBNext; E7500S, E7710S and E7730S). PolyA selection, cDNA synthesis and library generation were carried out using the Biomek 4000 Automated Laboratory Workstation (Beckman Coulter). Libraries were quantified using the Qubit Fluorometer and the Qubit dsDNA HS Kit (ThermoFisher Scientific; Q32854). Library fragment-length distributions were analysed using the Agilent TapeStation 4200 and the Agilent High Sensitivity D1000 ScreenTape Assay (Agilent; 5067-5584 and 5067-5585). Libraries were pooled in equimolar amounts and final library quantification performed using the KAPA Library Quantification Kit for Illumina (Roche; KK4824). The library pool was sequenced on the Illumina NextSeq500 using two NextSeq500 High Output 150 cycle kits (Illumina; 20024907), to generate around 30 million pairs of 75-bp paired-end reads per sample.

Low quality reads and adaptor sequences were removed from raw reads using default parameters of *skewer v0.2.2* software[66]. Filtered reads were mapped onto the reference genome (TAIR10) using the default parameters of *Hisat2 v2.1.0* software[67]. Only uniquely and correctly mapped reads were kept in bam files for the further analysis. Read counts for each gene are calculated using *featureCounts v1.6.0*. PCA was used to identify that the biological replicates are close enough to each other. *DESeq2 v1.24.0* was used to detect the differentially expressed genes for each comparison. and we used p-value < 0.5 as the DESeq thresholds. Gene ontology analysis of the differentially expressed genes was done using GOSeq v1.4.0 with default settings. Quantitative rtPCR was carried out as previously described[18] using a selection of known hypoxia up-regulated genes[68]. A list of oligonucleotide primers used for qRT-PCR is given in Supplementary Data 13. Gene Ontology (GO) representation analyses were carried out using the G:Profiler web resource (https://biit.cs.ut.ee/gprofiler/gost). Search for hypoxia-related cis-elements was carried out using the HRPE (GCCVCYGGTTTY) sequence[69] at MEME Suite 5.4.1 FIMO (https://meme-suite.org/meme/tools/fimo).

## Western blot and immunolocalization analyses of ERFVII stability

Western blots were carried out as previously described[26]. Primary antibodies used were anti-HA (Sigma, H3663-200UL; 1:1000 dilution), anti-FLAG (Sigma, F1804-200UG, 1:2000 dilution) and secondary antibody Goat anti-Mouse IgG1, HRP from Thermo Fisher Scientific, PA1 74421, (1:10000 dilution). Full scan uncropped Western blots are provided (Source Data 1). Immunolocalization of RAP2.3 was carried as described[70]. Four-day-old seedlings containing the *35S:RAP2.3³ˣᴴᴬ* transgene were treated for 3 h with 50 µM bortezomib or DMSO (control). Anti-HA primary antibody (Roche, Lewes, UK) and Alexa-Fluor-488-coupled anti rat secondary antibody (1:200 dilution)

(Molecular Probes, Carlsbad, CA) were used for detection. Seedlings were counter stained using Propidium Iodide and visualised using confocal microscopy.

## Generation of ubiquitin fusion technique construct

We developed a Ubiquitin Fusion Technique construct based on the plasmid pKP496 protein coding region [FLAG]DHFR[HA]-UBIQUITIN-[FLAG][43]. We replaced epitope tags to produce [FLAG]DHFR-UBIQUITIN-[3xHA] (pCD1). As described in[43] this provides a SacII restriction site downstream of the UBIQUITIN moiety that allows cloning in frame of test proteins with defined amino-terminal residues. In this case we cloned RAP2.3 starting at Cys[2], so that upon constitutive cleavage in vivo by deubiquitinating enzymes Cys2-RAP3.3[3xHA] is released from the pre-protein. This construct was subcloned into plant transformation gateway vector pH7wg2[71] providing ectopic constitutive expression from the 35S CaMV promoter to generate the construct pUFT:RAP2.3[3xHA]. Plants were transformed as previously described[59].

## Affinity purification of in vitro produced ERFVII

Protein coupled in vitro transcription/translation was carried out using a TnT® SP6 High-Yield Wheat Germ Protein Expression System (HYWGE) (Promega, USA) according to the manufacturer's recommendations. Reaction containing template pTNT-RAP2.3[3xHA]/RAP2.12[3xHA] HA-tagged constructs[4] contained 30 μl HYWGE with 20 μl of 300 ng/μl construct purified by QIAprep Spin Miniprep Kit (QIAgen, Crawley, UK)), reaction mixtures were incubated for 2 h at 25 C. Three reactions of TnT® SP6 High-Yield Wheat Germ Protein Expression System (HYWGE) (Promega, USA) were added separately to 50 μl of Pierce™ Anti-HA Magnetic Beads (Thermo Fisher, UK), and each topped up with 700 μl of freshly filtered (0.2 μm pore size) BupH Tris Buffered Saline Pack (25 mM Tris, 0.15 M NaCl; pH 7.2, Thermo Fisher, UK)). Samples were placed on an orbital rotor at 90 rpm. After 30 min samples were placed on a magnetic stand, the supernatant was removed, and beads were washed 3 times with 700 μl BupH Tris buffer (pH 7.2) with 0.1% Tween 20 (Fisher Bioreagents, UK). Each wash was for 2 min and was followed by magnetic precipitation of the beads. Beads were washed with Milli-Q water. Elution of the beads was achieved by adding 30 μl of SDS-PAGE Lane Marker Non-Reducing Sample Buffer (2X), (0.3 M Tris-HCl, pH 6.8, 5% SDS) (Thermo Fisher, UK) to the tube containing beads from the first reaction, followed by incubation at 96 °C for 10 min. Following precipitation, the supernatant was removed and transferred to the tube containing beads from the second reaction and the heating and precipitation steps repeated. Finally, the supernatant was transferred to the third tube, and after heating and precipitation, the supernatant was once again recovered and 1 μl of β-mercaptoethanol was added before loading onto an SDS-PAGE gel.

## Production of RAP2.3[3xHA] in H$_2$[18]O-containing medium

An aliquot of pTNT:RAP2.3[3xHA] plasmid (20 μl) was vacuum centrifuged until dry and resuspended in H$_2$[18]O (20 μl). This was then added to 30 μl of HYWGE to give a 40% H$_2$[18]O/ H$_2$[16]O solution (v/v). The cell free expression system was incubated and HA-tagged protein product purified as described above.

## SDS-PAGE and gel band excision

Samples from bead elution were loaded onto a 1.5 mm wide 12% gel SDS-PAGE gel. Gels were run for 20 min at 80 V until proteins passed from the stacking gel to the resolving gel, and then for 100–120 min at 120 V (for separation in the resolving gel). Staining was carried out in ~15 mL of Coomassie InstantBlue™ Protein Stain (Exepedeon, UK) for 1 h on an orbital shaker with rotation at 80 rpm. Gels were washed with Milli-Q water (×3) and left overnight for de-staining in Milli-Q water on an orbital shaker at 80 rpm. The band corresponding to purified protein of interest was excised with a sterile scalpel and cut into small pieces ready for in gel protein digestion.

## In gel protein digestion

Excised SDS-PAGE band slices were destained in 50 μl 50% acetonitrile (MeCN) (Fisher Scientific, Loughborough, UK) and 50% 100 mM ammonium bicarbonate (AmBic) (Sigma Aldrich). After 10 min incubation gel pieces were dehydrated by the addition of 450 μl of 100% MeCN with vortexing for 3 min (dehydration step). The supernatant was discarded, and gel pieces were treated with 50 μl of 10 mM dithiothreitol (DTT) (Sigma Aldrich) dissolved in 100 mM AmBic, and incubated at 55 °C for 30 min (DTT treatment). The dehydration step was repeated using 100% MeCN as above and gel slices were incubated with 50 μl of 10 mM iodoacetamide (Sigma Aldrich) in 100 mM AmBic for 30 min in the dark in order to block free Cys groups with a carbamidomethyl group. MeCN dehydration was repeated and additional DTT treatment was performed in order to consume excess IAA. After the final dehydration step gel slices were swollen with 50 μl 100 mM AmBic solution containing trypsin (0.4 ng μl⁻¹) and incubated at 37 °C overnight. After digestion samples were vortexed, centrifuged and supernatant was moved to a new 0.5 mL Eppendorf tube. Finally, samples were dried using a Savant DNA120 OP SpeedVac Concentrator (ThermoElectron, UK). Before LC-MS analysis, dried peptide samples were reconstituted in 15 μl of 70:30 H$_2$O:MeCN containing 0.1% formic acid and centrifuged (5000 RCF for 2 min) prior to transfer of the supernatant into tapered plastic LC vials (ThermoFisher, UK).

## LC-MS and LC-MS/MS analysis of protein digests

Digests were analysed on a Dionex U3000 nanoLC coupled to a ThermoFisher LTQ FT Ultra mass spectrometer (a hybrid linear ion trap-Fourier transform ion cyclotron resonance (FTICR) instrument equipped with a 7 Tesla magnet). Samples were loaded onto a C18 Pepmap300 cartridge (10 mm, 300 Å, 5 μm particle size) and separated using a Thermo Scientific C18 Pepmap300 column (150 mm × 75 μm, 300 Å, 5 μm particle size) attached to a 30 μm Picotip emitter (New Objective) in the nanoESI MS source. The mobile phases A and B consisted of 95:5 water:MeCN (v/v) and 5:95 water:MeCN (v/v), respectively, and both contained 0.1% formic acid. Samples (3 μL) were injected in load-trapping mode. Peptides were eluted using a 30 min linear gradient of mobile phase B from 0 to 55% followed by 5 min at 90% B and 15 min column re-equilibration. The LTQ FT Ultra mass spectrometer was equipped with a standard ThermoFisher nanoESI source through which a 1.8 kV voltage was applied to the Picotip emitter. The inlet capillary of the mass spectrometer was held at 275 °C with a tube lens value of 145 V.

For the identification of peptides/proteins the mass spectrometer was operated in data dependent acquisition (DDA) mode on the five most intense ions per each survey scan. Ions were isolated in the linear ion trap within a window of $m/z$ 8 and subjected to collision induced dissociation. Charge-state rejection of +1 ions was employed. The data were submitted to Search GUI database searches and results visualised in Peptide Shaker[72,73]. Parameters within Search GUI were as follows: MS Convert was used to generate.mgf data files using a 0.5 minimum peak spacing setting; X! Tandem was used for database searching with the following settings: Enzyme = Trypsin, Missed Cleavages = 2, Fixed Modifications = None, Variable Modifications = Carbamidomethylation of C, Oxidation of M, Precursor Tolerance = 0.5 Da, Product Tolerance = 0.5 Da, Fragment Ion Type = b/y, Charge State Range = +1 to +5. A FASTA decoy database file containing 250 sequences (forward and reverse) of common contaminant proteins together with the sequences of RAP2.3 and RAP2.12 was used for searches. For detailed examination of N-terminal modification of RAP2.3 and RAP2.12, data were analysed manually using Xcalibur software (ThermoFischer). Predicted

peptides of interest masses and fragmented ion masses were calculated with:

http://db.systemsbiology.net:8080/proteomicsToolkit/FragIonServlet.html

## Statistical analyses

No statistical methods were used in predetermining sample size and experiments were not randomized. The investigators were not blinded to allocation during experiments and outcome assessment. For experimental analyses at least three independent replicates with different original biological material are reported for each experiment and each experiment (excluding immunolocalizations) was repeated at least twice. All graphs were produced using Graphpad software (Version 9), which was also used to calculate standard deviation, $t$-test and one-way ANOVAs.

## Reporting summary

Further information on research design is available in the Nature Portfolio Reporting Summary linked to this article.

## Data availability

Data will to be shared upon request to Michael Holdsworth (michael.holdsworth@nottingham.ac.uk). The RNA-seq data generated in this study have been deposited in the GEO database under accession code GSE224694 (https://www.ncbi.nlm.nih.gov/geo/query/acc.cgi?acc=GSM7029623). The mass spectrometry proteomics data used in this study are available in the ProteomeXchange Consortium via the PRIDE[74] partner repository with the dataset identifier PXD039288. Source data are provided with this paper.

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

## Acknowledgements

We thank Professor Alex Varshavsky (Caltech, USA) for the kind gift of pKP496. We thank Jeddidiah Bellamy-Carter and James Lloyd for assistance with Mass Spectrometry analysis, and the DeepSeq Sequencing unit at Nottingham University for technical assistance and discussions associated with RNAseq experiments. The work was funded by a Leverhulme Trust Research Project Grant (RPG-2017-132) to M.J.H. and

was supported by the Biotechnology and Biological Sciences Research Council (grant number BB/R002428/1) to M.J.H. and N.J.O. A.Z was funded by a University of Nottingham joint Biosciences/Chemistry Ph.D. fellowship, and C.D. by the University of Nottingham Staff Development Fund. F.C. was funded by Sorbonne Université (Paris).

## Author contributions

M.J.H. and N.J.O. conceived the research with inputs from A.Z., G.S., C.D., L.D.C., R.S., C.S. and F.C.; M.J.H., N.J.O., A.Z., G.S., C.D., L.D.C., R.S., C.S., F.C., T.B., K.S. and B.A. designed the experiments, interpreted the data and carried out the research; M.J.H. and N.J.O. wrote the manuscript. M.J.H. agrees to serve as the author responsible for contact and ensures communication. All authors read, contributed to editing and approved the manuscript.

## Competing interests

The authors declare no competing interests.
