## [Peer Review File · Nature Communications]

ERFVII action and modulation through oxygen-sensing in *Arabidopsis thaliana*REVIEWER COMMENTS

Reviewer #1 (Remarks to the Author):

An oxygen sensing mechanism was previously described in *Arabidopsis thaliana*, and its discovery involved members of the same lab and others. A great deal of work has since been done to further elucidate the regulation of this oxygen sensing mechanism, as keenly summarized by the authors in the introduction. However, the authors of this manuscript highlight that key features of the associated PLANT CYSTEINE OXIDASE (PCO) N-degron pathway and Group VII ETHYLENE RESPONSE FACTOR (ERFVII) transcription factor substrates remain untested or unknown. They therefore set out to define the genetic and environmental components required for proteasome-dependent oxygen-regulated stability of ERF-VII through N-degron pathway. As written in the introduction, the authors set out to address the following gaps in the understanding of the known plant oxygen-sensing mechanism:

1. Define the role of ERFVIIs in regulating gene expression through the PCO N-degron pathway
2. Their influence in root responses to acute hypoxia
3. Their modulation through environmental and genetic factors
4. The influence of oxygen on the chemical nature of the amino-terminus

The presented data is of high quality and the authors have generated a valuable set of transgenic lines for the low oxygen community. However, this manuscript in part represents (an albeit important and more exhaustive) confirmation of previous findings. Moreover, this reviewer finds that the novel results require a more in-depth investigation of their significance and/or a better understanding of their regulation. As extrapolated on later they should: More deeply investigate the non-autonomous action of ERFVII. Provide a more complete understanding of the factors which control ERFVII proteolysis to induce hypoxia gene expression. Investigate further the role and mechanism by which N-terminal cysteine is modified to Cysteine sulfonic acid.

Major comments:

1. In Figure 3, the authors convincingly show that a majority of the *prt6* and hypoxia upregulated transcripts are ERFVII mediated, confirming the importance of this family for hypoxia-related gene expression. Interestingly, individual ERFVII genes have only limited or no capacity to induce hypoxia-related genes even when stabilized (Supplementary figure 1). I agree with the authors that this suggests that ERFVII members cannot act alone, although they do not investigate further how ERFVII might act interdependently. To understand this better, the authors should make combinations of individual ERFVIIs (with *prt6* and *erfvii* mutated) and measure root tip survival and hypoxia-related gene expression. Moreover, the authors should test if this interdependency could rely on protein-protein interaction between individual ERFVII members or interfamily transcriptional regulation.
2. The authors generated a Ubiquitin Fusion Technique (UFT) to investigate RAP2.3 stability in plants. Moreover, this has the benefit of circumventing the need for N-terminal methionine removal. By comparing this construct to a RAP2.3-3xHA, they showed that MetAP activity is not important for stabilization kinetics. Moreover, the authors neatly confirmed the requirement of the proteasome (via bortezomib), NO (via the CPTIO NO-scavenger), oxygen and iron for proteolysis of the ERFVII RAP2.3. They showed that BZ has the strongest effect on RAP2.3 stability, suggesting that degradation of RAP2.3 still occurs even in the absence of NO or oxygen, which is possibly N-degron pathway independent. To investigate if RAP2.3 is also degraded in an N-degron pathway independent-manner, the authors could use the UFT FLAG-DHFR-UB-x-RAP2.3-3xHA construct and investigate the relative stability of RAP2.3-3xHA compared to FLAG-DHR by WB when treated with BZ, CPTIO or hypoxia. Moreover, the authors could generate a UFT FLAG-DHFR-UB-x-RAP2.3-3xHA construct lacking the RAP2.3 N-terminus, to investigate if RAP2.3-specific proteolysis occurs that is independent of its N-terminus.

3. " Stability was higher in the presence of the NO scavenger cPTIO than under reduced oxygen conditions." . What was the oxygen concentration used here? Is enough oxygen still present to promote degradation and could the authors test if cPTIO scavenges all NO?
4. The authors observe cysteine sulfonic acid as a new modification in vivo, while cysteine sulfinic acid was only observed so far observed in vitro. This is a novel and exciting finding, which opens the door to the discovery of a new potentially oxygen-dependent enzyme(s) that regulates the pathway as also mentioned by the authors. However, the role and regulation of the sulfonic acid modification remains unknown in this manuscript.
5. Figure 1e. The weight of rosettes and roots show a high degree of variation even under control conditions, i.e. some plants have triple the weight as others. If this is the natural variation under these growth conditions, the tolerance experiment should be repeated with more replicates to make this assay more reliable and to be able to compare statistical differences between the wildtype and ervii mutant.
6. Figure 1b. prt6 impressively enhanced root tip survival. Moreover, the prt6 ervii showed that most of the improved tolerance mediated by prt6 is ERFVII-dependent. However, the ervii mutant showed no significant difference compared to the wildtype, raising doubts about the importance of ERFVII in root tip survival when ERFVII are not stabilized prior to the stress (as in prt6). To strengthen the relevance of this mechanism, could the authors investigate tolerance of ervii at different time-points of hypoxia treatment, or by gradually decreasing oxygen levels to mimic natural occurring hypoxia conditions, which would pre-stabilize ERFVII in wildtype plants?
7. Supplementary figure 1c. These expression analysis were performed using whole seedlings. Could the authors compare expression of ADH1, PCO1 and SUS4 genes in root tips or in entire roots to better correlate this to root tip survival in the different transgenic lines?
8. Expression of MA versions of the ERFVII led to an almost comparable root tip survival as prt6 (figure 1b), but this is not accompanied by increased PCO1, SUS4 or ADH1 expression (supplementary figure 1c). If upregulation of these genes in MA-ERFVII is not the cause of enhanced tolerance, then what do the authors propose is the mechanism for enhanced tolerance here?

Minor comments:

1. Supplementary figure 1c: the authors write that "individually RAP2.2 ad RAP2.3 stabilised in the prt6 background did enhance expression of the three core hypoxia genes analysed (Supplementary Fig. 1c)."

The data show increased upregulation of ADH1 by RAP2.2 and RAP2.12, and upregulation of PCO1 by RAP2.2, RAP2.3 and RAP2.12. SUS4 was only induced by Rap2.12. Please rephrase text to represent this.

2. However, whereas C2 A-ERFVII could not enhance hypoxia-related gene expression individually RAP2.2 ad RAP2.3 stabilised in the prt6 background did enhance expression of the three core hypoxia genes analysed (Supplementary Fig. 1c).

"ad" should be changed to "and"

Reviewer #2 (Remarks to the Author):

In this manuscript Zubrycka et al. present a series of evidences on the action of ERFVII on the hypoxic response in Arabidopsis

Overall the manuscript contains a lot of interesting information, although somehow apparently not fully related with each other.

The first set of experiments, reported in Fig. 1, is extremely interesting. The authors demonstrate that the five ERFVII do not act redundantly to protect the root tip from the consequences of hypoxia. Neither of the five ERFVII alone is indeed able to restore the strong tolerance in the prt6 mutant. Producing the lines used in this experiment have been

for sure a very long work and the authors deserve kudos for this. Remarkably, expression of the single ERFVII in this MA version confers hypoxia tolerance. RAP2.2 seems to be the ERFVII required during germination. All these information suggest that each single ERFVII may play a specific role in either specific tissues or developmental stages.

Figure 2 shows that ERFVII influence on root structure in soil. This was achieved using a sophisticated X-ray computed micro-tomography approach. A slightly reduced growth was also observed in erfVII roots and rosettes under non-stressed conditions, suggesting a new role of ERFVIIs in plant growth and development. This is indeed of interest and in agreement with a previous article showing a growth deficit in mutants affected in the fermentative pathway (adh and pdc: Ventura et al. Sci Rep 2020 Oct 7;10(1):16669).

The transcriptomic analysis of Col vs prt6 vs prt6-erfvii (Fig.3) is very interesting and may reveal additional components that are required for the response to hypoxia, beyond ERFVII but within the targets of PRT6.

The timing of ERFVII stabilization (as well as the role of Fe) are of interest. I agree with the authors that stabilization of newly transcribed ERFVII is more than sufficient for explaining the fast induction of HRGs without the need of including the possible release of ERFVII from ACBPs in the membrane.

Finally, the authors analyze the chemical structure at the aminoterminal of RAP2.3 and RAP2.12. Three oxygen atoms are required for the oxidation of the Cys residue and the authors provide evidence that all three atoms are derived from O₂, not water.

My feeling after reading this manuscript is that it contains a very large amount of new information on the dynamics and molecular mechanisms behind ERFVII as activators of the hypoxic response at the transcriptional level. Although the five sections of the manuscript are not necessarily sequentially interconnected the quality of this work is very high.

Reviewer #3 (Remarks to the Author):

Overall, this is a very well designed study on oxygen sensing in Arabidopsis, and the role of the ERFVII family of transcription factors. It is a very large and wide ranging set of experiments that uncovers some novel biological information, showing that ERFVII family members are essential for root development under oxygen limited environments, and are also involved in activation of root hypoxaemia tolerance. This builds on previous work in the area, and represents a significant advance in the field. I think this is a very nice piece of work, but some revisions are necessary. Note that I have been asked to focus specifically on the proteomics aspects of this study.

Transcriptomic data seems to be discussed at a fairly big picture superficial level, with little consideration of individual transcript changes. I realise that is a lot of data to deal with, but there must be at least a few things that could be highlighted.

The use of 3-D micro-CT imaging to look at root structures in situ is an excellent idea, and shows really nice data as a result. This is a new technique to me, and certainly gives higher quality data than the use of root boxes.

I might have missed something here, but I am a little confused by the use of the term "stabilised" throughout. For example, in the first paragraph on page 8 talking about genetic adaptation to altitude. The transgene is referred to as being either more abundant or more stable. Surely the converse of more abundant is less abundant, and the level in between is unchanged, or something like that. Similarly on page 10, the exact meaning of the "individually stabilised ERFVII proteins" was not totally clear to me.

It would be a good idea to check that the term 'stabilised' is actually used as intended throughout, and not inadvertently used on occasions when the intent was to refer to individual gene expression.

The use of the LC-MS techniques to confirm the identity of the two bands from the HA pulldown seems like a reasonable approach. The mass spectrometric instrumentation, and software used for peptide to spectrum matching, both seem to be fairly 'old school' but are still fit for purpose in this instance. These are more than sufficient to perform high quality experiments with the aim of identifying specific peptides and modifications.

As far as I understand, Search GUI is a high level wrapper which incorporates several different peptide to spectrum matching algorithms. The methods should definitely state which algorithms were actually used, and with which parameters. It is also standard practice to describe the Fasta protein sequence file used to search spectra against, including how many entries it contains. The raw data is available from the Pride database, which is good, but the level of detail provided in the methods currently is not sufficient to allow a reader to repeat those peptide to spectrum matching experiments.

The one caveat I would place on the data is that only cutting out specific bands means you might be missing other proteins which are present at much lower abundance in other areas of the gel. This is a big issue in protein complex purification, with complexes of differing abundance co-purifying, but in this case I don't really think it is likely to be too much of a problem.

Similarly the use of ^{18}O water to determine the source of oxygen atoms incorporated into the cysteine sulphonic acid residues is a really nice idea, and a good application of the technique. The data produced support the hypothesis that the third oxygen atom involved is introduced via an as yet unknown enzyme, which is an intriguing finding, and definitely worth following up.

Minor errors noted:
RAP2.2 ad RAP2.3

I waive my right to anonymity.

Regards

Paul A. Haynes

Response to reviewers: Nature Communications manuscript NCOMMS-23-05470

All responses in red
In the revised manuscript all changed text is in red

REVIEWER COMMENTS

Reviewer #1 (Remarks to the Author):

An oxygen sensing mechanism was previously described in *Arabidopsis thaliana*, and its discovery involved members of the same lab and others. A great deal of work has since been done to further elucidate the regulation of this oxygen sensing mechanism, as keenly summarized by the authors in the introduction. However, the authors of this manuscript highlight that key features of the associated PLANT CYSTEINE OXIDASE (PCO) N-degron pathway and Group VII ETHYLENE RESPONSE FACTOR (ERFVII) transcription factor substrates remain untested or unknown. They therefore set out to define the genetic and environmental components required for proteasome-dependent oxygen-regulated stability of ERF-VII through N-degron pathway. As written in the introduction, the authors set out to address the following gaps in the understanding of the known plant oxygen-sensing mechanism:

1. Define the role of ERFVIIs in regulating gene expression through the PCO N-degron pathway
2. Their influence in root responses to acute hypoxia
3. Their modulation through environmental and genetic factors
4. The influence of oxygen on the chemical nature of the amino-terminus

The presented data is of high quality and the authors have generated a valuable set of transgenic lines for the low oxygen community.

We thank the referee for highlighting the value of the genetic resources we generated in this study. Mutant lines have already been deposited at the International Arabidopsis Stock Centre (NASC) as part of the submission to Nature Communications.

However, this manuscript in part represents (an albeit important and more exhaustive) confirmation of previous findings.

We do not agree with this assertion by the referee. Work in this manuscript certainly is not “confirmation of previous findings”. We provide important new data defining the overall importance of ERFVIIs as transducers of oxygen-sensing in root survival in the soil, global regulation of gene expression in relation to the PRT6 N-degron pathway, and exact biochemical changes occurring at the amino-terminus as a result of oxygen sensing. None of these findings have been reported previously and all add significantly to our understanding of the plant oxygen-sensing system. Ref #2 states that “it contains a very large amount of new information” and Ref #3 “This builds on previous work in the area, and represents a significant advance in the field”.

Moreover, this reviewer finds that the novel results require a more in-depth investigation of their significance and/or a better understanding of their regulation. As extrapolated on later they should: More deeply investigate the non-autonomous action of ERFVII. Provide a more complete understanding of the factors which control ERFVII proteolysis to induce hypoxia gene expression. Investigate further the role and mechanism by which N-terminal cysteine is modified to Cysteine sulfonic acid.

We thank the referee for their very detailed review of our manuscript, but respectfully strongly disagree with the suggestion of extended in depth investigations. The very large and diverse number of additional experiments requested by the referee will not alter the conclusions of the paper nor add to their validity and are out of scope for the hypotheses investigated in the manuscript. In that case we do not consider their addition necessary. The work in the different areas proposed by the referee would find their natural place in follow on studies subsequent to this manuscript focussed on specific aspects of ERFVII biology. We discuss individual requests below, and also point out that neither of the other two referees requested similar experiments, but did highlight the importance and depth of the presented experimental results.

Major comments:

1. In Figure 3, the authors convincingly show that a majority of the prt6 and hypoxia upregulated transcripts are ERFVII mediated, confirming the importance of this family for hypoxia-related gene expression. Interestingly, individual ERFVII genes have only limited or no capacity to induce hypoxia-related genes even when stabilized (Supplementary figure 1). I agree with the authors that this suggests that ERFVII members cannot act alone, although they do not investigate further how ERFVII might act Interdependently. To understand this better, the authors should make combinations of individual ERFVIIs (with prt6 and erfvi mutated) and measure root tip survival and hypoxia-related gene expression.

This request would not change the conclusion of presented data that ERFVIIs act non-autonomously, but would consume a huge amount of human and physical resources (and time to generate many of the new genetic combinations required). For 2 possible states (WT and mutant) of six genes we would have to generate and analyse 64 different genotype combinations. As we show that each ERFVII is capable in the stabilised form of enhancing hypoxia survival we could not prioritise specific combinations. Previous studies implicated the constitutive ERFVIIs RAP2.12 RAP2.2 and RAP2.3 as “more important”, but our results show that all five ERFVIIs have the same potency when stabilised, and all act non-autonomously. Therefore, we consider this request more suited to follow on studies focussed on individual ERFVIIs.

Moreover, the authors should test if this interdependency could rely on protein-protein interaction between individual ERFVII members or interfamily transcriptional regulation.

As discussed for the previous point, we do not consider the very large amount of experimental work requested necessary as it would not change the conclusions of the manuscript. Results from suggested experiments would only be tangentially related to the focus of the hypotheses investigated in the manuscript, that address the link between ERFVIIs and the oxygen-sensing system. We consider this request more suited to follow on studies focussed on individual ERFVIIs.

2. The authors generated a Ubiquitin Fusion Technique (UFT) to investigate RAP2.3 stability in plants. Moreover, this has the benefit of circumventing the need for N-terminal methionine removal. By comparing this construct to a RAP2.3-3xHA, they showed that MetAP activity is not important for stabilization kinetics. Moreover, the authors neatly confirmed the requirement of the proteasome (via bortezomib), NO (via the CPTIO NO-scavenger), oxygen and iron for proteolysis of the ERFVII RAP2.3. They showed that BZ has the strongest effect on RAP2.3 stability, suggesting that degradation of RAP2.3 still occurs even in the absence of NO or oxygen, which is possibly N-degron pathway independent. To investigate if RAP2.3 is also degraded in an N-degron pathway independent-manner, the authors could use the UFT FLAG-DHFR-UB-x-RAP2.3-3xHA construct and investigate the relative stability of

RAP2.3-3xHA compared to FLAG-DHR by WB when treated with BZ, CPTIO or hypoxia. Moreover, the authors could generate a UFT FLAG-DHFR-UB-x-RAP2.3-3xHA construct lacking the RAP2.3 N-terminus, to investigate if RAP2.3-specific proteolysis occurs that is independent of its N-terminus.

As we already showed that +BZ treatment leads to much greater stability than in the *prt6* mutant, we have already shown that RAP2.3-specific proteolysis occurs that is independent of its N-terminus. In fact it was already shown by several other published studies (eg; Papdi, C. et al. The low oxygen, oxidative and osmotic stress responses synergistically act through the ethylene response factor VII genes RAP2.12, RAP2.2 and RAP2.3. *Plant J.* 82, 772-784 (2015)) as detailed in the manuscript introduction/discussion, that other E3 ligases influence ERFVII stability independent of the N-terminus. These regulators are not part of the oxygen-sensing system and therefore not of importance to the hypotheses being tested in this manuscript, linking ERFVII to the oxygen-sensing system.

3. "Stability was higher in the presence of the NO scavenger cPTIO than under reduced oxygen conditions." . What was the oxygen concentration used here? Is enough oxygen still present to promote degradation and could the authors test if cPTIO scavenges all NO?

This question is not possible to answer. At the concentration of cPTIO used in our experiments (similar to that used in many other studies employing cPTIO to scavenge NO) it is likely that almost all NO is scavenged but it would be impossible to actually prove that ALL intracellular NO was scavenged. Similarly, it is not possible to prove that ALL intracellular oxygen is removed. However, submerged seedlings stabilise RAP2.3 rapidly by 10 minutes at levels that remains relatively constant still after 60 minutes, leading to the conclusion that all *available* oxygen is removed very fast (otherwise RAP2.3 would continue to accumulate through the time course).

The oxygen concentration used in hypoxia was not explicitly mentioned in the text, and so has been added to both the results and methods sections (<0.5% O₂).

4. The authors observe cysteine sulfonic acid as a new modification in vivo, while cysteine sulfinic acid was only observed so far observed in vitro. This is a novel and exciting finding, which opens the door to the discovery of a new potentially oxygen-dependent enzyme(s) that regulates the pathway as also mentioned by the authors. However, the role and regulation of the sulfonic acid modification remains unknown in this manuscript.

We are happy that the referee agrees that this is a very important finding, as it changes the paradigm for how the oxygen-sensing system works. Follow on studies will surely define the role and regulation of this modification. In the discussion we provide some ideas for area of investigation and possible enzymatic activity.

5. Figure 1e. The weight of rosettes and roots show a high degree of variation even under control conditions, i.e. some plants have triple the weight as others. If this is the natural variation under these growth conditions, the tolerance experiment should be repeated with more replicates to make this assay more reliable and to be able to compare statistical differences between the wildtype and *ervii* mutant.

In Fig. 1e plants were grown from seed in natural soil collected from a local farm. It is not unsurprising then that growth of plants will be highly heterogenous compared to defined soil used in "normal" experimental procedures for growing plants transferred to soil as seedlings. It is unclear how repeating the experiment more times will make results more reliable. Clear

statistically significant differences are observed between treatments, these are detailed in the figure legend.

6. Figure 1b. *prt6* impressively enhanced root tip survival. Moreover, the *prt6 erfVII* showed that most of the improved tolerance mediated by *prt6* is ERFVII-dependent. However, the *erfVII* mutant showed no significant difference compared to the wildtype, raising doubts about the importance of ERFVII in root tip survival when ERFVII are not stabilized prior to the stress (as in *prt6*). To strengthen the relevance of this mechanism, could the authors investigate tolerance of *erfVII* at different time-points of hypoxia treatment, or by gradually decreasing oxygen levels to mimic natural occurring hypoxia conditions, which would pre-stabilize ERFVII in wildtype plants?

The referee is correct in stating that these results show no statistical difference between WT and *erfVII* mutant for root hypoxia tolerance. However, the aim of the experiment was to assess the contributions of ERFVIIs to hypoxia tolerance within the PRT6 N-degron pathway by comparing tolerances of mutant combinations to that of *prt6* (all substrates stabilised) and *prt6 erfVII* (all substrates except ERFVIIs stabilised). Thus, *prt6* vs *prt6 erfVII* allows an assessment of ERFVII function, whilst *erfVII* vs *prt6 erfVII* allows an assessment of NON ERFVII PRT6 substrates. Lack of difference in phenotype between WT and *erfVII* only implies that this assay is not sensitive at the bottom end to pick up potential differences between these two genotypes. This is not required to answer the question proposed.

In developing this assay, we tested both level of ambient oxygen and time of assay (see unpublished data in figure below). As can be seen tolerance of *prt6* is most apparent when oxygen levels become very low, and after 4 hours of treatment. This assay has also been used in several other published studies by us and other groups (see response to point 8).

Influence of time in hypoxia (<0.5% O₂) on root tip survival

Influence of different ambient O₂ levels on root tip survival

7. Supplementary figure 1c. These expression analysis were performed using whole seedlings. Could the authors compare expression of ADH1, PCO1 and SUS4 genes in root tips or in entire roots to better correlate this to root tip survival in the different transgenic lines?

The reported experiments are not directly related to the root top assay because they are carried out in seedlings in normoxia. The aim of the experiments was to define the ability of

stabilised ERFVIIIs (either genetically through *prt6* removal or by altering the N terminus) to constitutively activate expression of core hypoxia genes.

This request by the referee would not change the conclusion of presented data as described above but would consume a huge amount of human and physical resources (including time generating large amount of seed material to obtain enough roots for QrtPCR analysis). Statistically significant differences in gene expression are observed in the presented data using whole seedlings. In that case we do not consider it necessary to carry out further experiments. A future study investigating the exact role of ERFVIIIs in gene expression during root tip survival could address these points. We highlight the recent paper by Liu *et al.* that investigated the mechanism of root tip survival of hypoxia (see answer to point 8).

8. Expression of MA versions of the ERFVII led to an almost comparable root tip survival as *prt6* (figure 1b), but this is not accompanied by increased PCO1, SUS4 or ADH1 expression (supplementary figure 1c). If upregulation of these genes in MA-ERFVII is not the cause of enhanced tolerance, then what do the authors propose is the mechanism for enhanced tolerance here?

The root tip survival assay has been used in several published studies over the last 5 years (Gibbs et al Nat Comms 2018, Hartman et al Nat Comms 2019, Barreto et al Current Biol 2022). Last year Liu *et al* (Liu *et al.* Ethylene augments root hypoxia tolerance via growth cessation and reactive oxygen species amelioration. Plant Physiol. (2022)) showed using this method that the most likely mechanism involves ethylene regulation of enhanced hypoxia responses, amelioration of ROS and reduction of root growth.

Minor comments:

1. Supplementary figure 1c: the authors write that “individually RAP2.2 ad RAP2.3 stabilised in the *prt6* background did enhance expression of the three core hypoxia genes analysed (Supplementary Fig. 1c).”

The data show increased upregulation of ADH1 by RAP2.2 and RAP2.12, and upregulation of PCO1 by RAP2.2, RAP2.3 and RAP2.12. SUS4 was only induced by Rap2.12. Please rephrase text to represent this.

We have rephrased the text to make it more understandable:

“C²A-ERFVIIIs could not enhance hypoxia-related gene expression (with the exception of *SUS4* for MA-RAP2.12 (Supplementary Fig. 1c). Both RAP2.2 and RAP2.3 when individually stabilised by the *prt6* background did enhance expression of the three core hypoxia transcripts analysed, and RAP2.3 one transcript (Supplementary Fig. 1c).”

2. However, whereas C2 A-ERFVIIIs could not enhance hypoxia-related gene expression individually RAP2.2 ad RAP2.3 stabilised in the *prt6* background did enhance expression of the three core hypoxia genes analysed (Supplementary Fig. 1c).

“ad” should be changed to “and”

We thank the referee for pointing out this typographical error, that has been corrected.

Reviewer #2 (Remarks to the Author):

In this manuscript Zubrycka et al. present a series of evidences on the action of ERFVII on the hypoxic response in Arabidopsis

Overall the manuscript contains a lot of interesting information, although somehow

apparently not fully related with each other.

The first set of experiments, reported in Fig. 1, is extremely interesting. The authors demonstrate that the five ERFVII do not act redundantly to protect the root tip from the consequences of hypoxia. Neither of the five ERFVIV alone is indeed able to restore the strong tolerance in the prt6 mutant. Producing the lines used in this experiment have been for sure a very long work and the authors deserve kudos for this.

As stated for Ref#1; We thank the referee for highlighting the value of the genetic resources we generated in this study. Mutant lines have already been deposited at the International Arabidopsis Stock Centre (NASC) as part of the submission to Nature Communications.

Remarkably, expression of the single ERFVII in this MA version confers hypoxia tolerance. RAP2.2 seems to be the ERFVII required during germination. All these information suggest that each single ERFVII may play a specific role in either specific tissues or developmental stages.

Figure 2 shows that ERFVII influence on root structure in soil. This was achieved using a sophisticated X-ray computed micro-tomography approach. A slightly reduced growth was also observed in erfVII roots and rosettes under non-stressed conditions, suggesting a new role of ERFVIIs in plant growth and development This is indeed of interest and in agreement with a previous article showing a growth deficit in mutants affected in the fermentative pathway (adh and pdc: Ventura et al. Sci Rep 2020 Oct 7;10(1):16669).

We thank the referee for this useful comment. We added a sentence to the discussion to highlight the relevance of our finding in relation to this paper:

“This observation is in agreement with the recent finding that lack of functional *ADH1* or *PDC1* leads to a growth penalty under normal (aerobic) growth conditions ⁴⁸.”

The transcriptomic analysis of Col vs prt6 vs prt6-erfvii (Fig.3) is very interesting and may reveal additional components that are required for the response to hypoxia, beyond ERFVII but within the targets of PRT6.

We thank the referee for highlighting this important point, that the described RNAseq transcriptomics work will allow an understanding of ERFVII function within their regulation by PRT6.

The timing of ERFVII stabilization (as well as the role of Fe) are of interest. I agree with the authors that stabilization of newly transcribed ERFVII is more than sufficient for explaining the fast induction of HRGs without the need of including the possible release of ERFVII from ACBPs in the membrane.

We thank the referee for highlighting this point. We agree that the community needs a discussion about the importance and relevance of possible release of ERFVII from ACBPs in the membrane. Hopefully our results will stimulate such a discussion.

Finally, the authors analyze the chemical structure at the aminotermius of RAP2.3 and RAP2.12, Three oxygen atoms are required for the oxidation of the Cys residue and the authors provide evidence that all three atoms are derived from O₂, not water. My feeling after reading this manuscript is that it contains a very large amount of new information on the dynamics and molecular mechanisms behind ERFVII as activators of the hypoxic response at the transcriptional level. Although the five sections of the manuscript are not necessarily sequentially interconnected the quality of this work is very high.

It is true that the sections of the manuscript are not necessarily sequential, but they all address key missing information about the plant oxygen-sensing system, and the results obtained provide a holistic understanding of the role of ERFVII as transducers of response to reduced oxygen through the oxygen-sensing system.

Reviewer #3 (Remarks to the Author):

Overall, this is a very well designed study on oxygen sensing in Arabidopsis, and the role of the ERFVII family of transcription factors. It is a very large and wide ranging set of experiments that uncovers some novel biological information, showing that ERFVII family members are essential for root development under oxygen Limited saw environments, and are also involved in activation of root hypoxaemia tolerance. This builds on previous work in the area, and represents a significant advance in the field. I think this is a very nice piece of work, but some revisions are necessary. Note that I have been asked to focus specifically on the proteomics aspects of this study.

Transcriptomic data seems to be discussed at a fairly big picture superficial level, with little consideration of individual transcript changes. I realise that is a lot of data to deal with, but there must be at least a few things that could be highlighted.

We are unclear what the referee would like highlighting? Information about individual core hypoxia genes is given in Figure 3d. There is a large amount of data defining the relationships between ERFVII/PRT6 regulated genes/hypoxia regulated genes and cis-element representation in the Supplementary tables. Our aim in this manuscript was to provide a global analysis of how ERFVII control transcript abundance through the PRT6 branch of the N-degron pathways. This will provide a reference for other researchers in the field for further analysis. As our data was produced by RNAseq, it would be possible for readers to directly find the number of RNA molecules for any transcript in the genotypes we analysed.

The use of 3-D micro-CT imaging to look at root structures in situ is an excellent idea, and shows really nice data as a result. This is a new technique to me, and certainly gives higher quality data than the use of root boxes.

We thank the referee for this observation. We agree that analysis of root growth and survival in situ in the natural soil environment represents a step-change in ability to understand below ground plant biology. In our case to show the key role of ERFVII in maintaining root growth and development in waterlogged soil.

I might have missed something here, but I am a little confused by the use of the term “stabilised” throughout. For example, in the first paragraph on page 8 talking about genetic adaptation to altitude. The transgene is referred to as being either more abundant or more stable. Surely the converse of more abundant is less abundant, and the level in between is unchanged, or something like that. Similarly on page 10, the exact meaning of the “individually stabilised ERFVII proteins” was not totally clear to me.

It would be a good idea to check that the term 'stabilised' is actually used as intended throughout, and not inadvertently used on occasions when the intent was to refer to individual gene expression.

We thank the referee for pointing out this potential for misunderstanding of terminology. Page 8: The sentence was split and changed to:

“In etiolated seedlings at pO_2 21 kPa RAP2.3^{3xHA} was more abundant in Col-0 than Sha. However, under lower ambient oxygen (15%, equivalent to 14.7 kPa) RAP2.3^{3xHA} was as abundant in high altitude accession Sha as in low altitude Col-0 (Fig. 4h).”

Throughout the manuscript we have altered stabilised/stabilisation to accumulation/abundance/degradation etc where appropriate.

The use of the LC-MS techniques to confirm the identity of the two bands from the HA pulldown seems like a reasonable approach. The mass spectrometric instrumentation, and software used for peptide to spectrum matching, both seem to be fairly 'old school' but are still fit for purpose in this instance. These are more than sufficient to perform high quality experiments with the aim of identifying specific peptides and modifications.

As far as I understand, Search GUI is a high level wrapper which incorporates several different peptide to spectrum matching algorithms. The methods should definitely state which algorithms were actually used, and with which parameters. It is also standard practice to describe the Fasta protein sequence file used to search spectra against, including how many entries it contains. The raw data is available from the Pride database, which is good, but the level of detail provided in the methods currently is not sufficient to allow a reader to repeat those peptide to spectrum matching experiments.

We are grateful to the reviewer for flagging the need to include more detail for the Search GUI parameters. We have added the following information to the experimental section of the manuscript:

“Parameters within Search GUI were as follows: MS Convert was used to generate .mgf data files using a 0.5 minimum peak spacing setting; X! Tandem was used for database searching with the following settings: Enzyme = Trypsin, Missed Cleavages = 2, Fixed Modifications = None, Variable Modifications = Carbamidomethylation of C, Oxidation of M, Precursor Tolerance = 0.5 Da, Product Tolerance = 0.5 Da, Fragment Ion Type = b/y, Charge State Range = +1 to +5. A FASTA decoy database file containing 250 sequences (forward and reverse) of common contaminant proteins together with the sequences of RAP2.3 and RAP2.12 was used for searches.”

FASTA sequences for both RAP2.12 and RAP2.3 have been added to the manuscript at the beginning of the Supplementary Data 1 file. The sequences are also included in Figure S5b, with the N-terminal Arginylation shown.

The one caveat I would place on the data is that only cutting out specific bands means you might be missing other proteins which are present at much lower abundance in other areas of the gel. This is a big issue in protein complex purification, with complexes of differing abundance co-purifying, but in this case I don't really think it is likely to be too much of a problem.

In this study we were only interested in understanding the status of the amino-terminus of the ERFVIs studied and so we only cut out the major band corresponding to the ERFVII protein in the gel.

Similarly the use of 18-O water to determine the source of oxygen atoms incorporated into the cysteine sulphonic acid residues is a really nice idea, and a good application of the technique. The data produced support the hypothesis that the third oxygen atom involved is introduced via an as yet unknown enzyme, which is an intriguing finding, and definitely worth following up.

We thank the referee for agreeing that our data support the idea of an unknown enzyme in the oxygen-sensing system, and that this work will promote important and interesting follow-on research.

Minor errors noted:
RAP2.2 ad RAP2.3

We thank the referee for pointing out this typographical error, that has been corrected.

I waive my right to anonymity.

Regards

Paul A. Haynes

VIEWERS' COMMENTS

Reviewer #1 (Remarks to the Author):

The authors have addressed points 3, 6, and 8 satisfactorily. However, my concerns regarding other aspects have not been adequately resolved. I want to emphasize that this work contains intriguing and novel findings. However, as elaborated on in my original review, further investigation is needed to substantiate the authors' claims and provide a more holistic understanding of these findings. While the authors have valid reasons in terms of resource and time constraints for not conducting a more in-depth investigation at this stage, I respectfully disagree with their assertion that this does not impact the paper's conclusions or its validity.

Reviewer #2 (Remarks to the Author):

The authors have edited their manuscript in line with my comments. I do not have other comments: this manuscript will provide the community with answers but also very valuable questions about unresolved aspects of oxygen sensing in plants.

Reviewer #3 (Remarks to the Author):

I am mostly satisfied with the response of the authors to the minor concerns I have raised from the original manuscript, apart from one thing which seems to need further explanation, which is that I still think the treatment of the transcriptomic data is a little superficial. I agree that figure 3b presents some information on enriched functional categories, and figure 3-D refers to fold change of a predefined set of 49 previously known hypoxia induced genes. However, I was hoping to see some new information on specific transcripts/genes altered in abundance in this study, and I can't really seem to find that here. It might be somewhere in the 12 supplementary tables, many of which have thousands of lines of data each, but that is why I thought perhaps highlighting some of the most relevant new discoveries in terms of genes altered in abundance in these various comparisons might be a good idea. I'm not sure if I am even meant to comment on the issues raised by the other reviewers, but I figured I might as well say something. I agree with the authors that the very large amount of extra experiments requested by the first reviewer don't seem like they would really add much new biological information to the manuscript. I also had a similar question to reviewer two regarding the apparent lack of difference between WT and erf vii mutant in terms of root hypoxia tolerance, but I think that has been successfully addressed.

Paul A. Haynes

Nature Communications manuscript NCOMMS-23-05470A response to REVIEWER COMMENTS

Response to referee comments in red

REVIEWERS' COMMENTS

Reviewer #1 (Remarks to the Author):

The authors have addressed points 3, 6, and 8 satisfactorily. However, my concerns regarding other aspects have not been adequately resolved. I want to emphasize that this work contains intriguing and novel findings. However, as elaborated on in my original review, further investigation is needed to substantiate the authors' claims and provide a more holistic understanding of these findings. While the authors have valid reasons in terms of resource and time constraints for not conducting a more in-depth investigation at this stage, I respectfully disagree with their assertion that this does not impact the paper's conclusions or its validity.

> We thank the referee for their comments and understanding our valid reasons.

Reviewer #2 (Remarks to the Author):

The authors have edited their manuscript in line with my comments. I do not have other comments: this manuscript will provide the community with answers but also very valuable questions about unresolved aspects of oxygens sensing in plants.

> We thank the referee for their comments and agree that we consider our manuscript important for providing answers but also valuable questions for future investigation about unresolved aspects of oxygens sensing in plants.

Reviewer #3 (Remarks to the Author):

I am mostly satisfied with the response of the authors to the minor concerns I have raised from the original manuscript, apart from one thing which seems to need further explanation, which is that I still think the treatment of the transcriptomic data is a little superficial. I agree that figure 3b presents some information on enriched functional categories, and figure 3-D refers to fold change of a predefined set of 49 previously known hypoxia induced genes. However, I was hoping to see some new information on specific transcripts/genes altered in abundance in this study, and I can't really seem to find that here. It might be somewhere in the 12 supplementary tables, many of which have thousands of lines of data each, but that is why I thought perhaps highlighting some of the most relevant new discoveries in terms of genes altered in abundance in these various comparisons might be a good idea. I'm not sure if I am even meant to comment on the issues raised by the other reviewers, but I figured I might as well say something. I agree with the authors that the very large amount of extra experiments requested by the first reviewer don't seem like they would really add much new biological information to the manuscript. I also had a similar question to reviewer two regarding the apparent lack of difference between WT and erf vii mutant in terms of root hypoxia tolerance, but I think that has been successfully addressed.

> We thank the referee for their comments. Information about differential expression of individual genes derived from RNAseq is available in Supplementary Tables 1-3.